# BRG1 HSA domain interactions with BCL7 proteins are critical for remodeling and gene expression

Nicholas Dietrich[1], Kevin Trotter[1], James M Ward[2], Trevor K Archer[1]

The SWI/SNF complex remodels chromatin in an ATP-dependent manner through the subunits BRG1 and BRM. Chromatin remodeling alters nucleosome structure to change gene expression; however, aberrant remodeling can result in cancer. We identified BCL7 proteins as critical SWI/SNF members that drive BRG1-dependent gene expression changes. BCL7s have been implicated in B-cell lymphoma, but characterization of their functional role within the SWI/SNF complex has been limited. This study implicates their function alongside BRG1 to drive large-scale changes in gene expression. Mechanistically, the BCL7 proteins bind to the HSA domain of BRG1 and require this domain for binding to chromatin. BRG1 proteins without the HSA domain fail to interact with the BCL7 proteins and have severely reduced chromatin remodeling activity. These results link the HSA domain and the formation of a functional SWI/SNF remodeling complex through the interaction with BCL7 proteins. These data highlight the importance of correct formation of the SWI/SNF complex to drive critical biological functions, as losses of individual accessory members or protein domains can cause loss of complex function.

## Introduction

ATP-dependent chromatin remodeling complexes, such as the SWItch/Sucrose Non-Fermentable (SWI/SNF) complex, perform a critical biological function in altering the contacts between histones and DNA in chromatin (Wu et al, 2017; Dietrich et al, 2020). This enzymatic function, via chromatin remodeling, drives a plethora of biological roles including organismal development, cellular differentiation, cell cycle progression, DNA repair, and transcriptional regulation (de la Serna et al, 2006; Ho & Crabtree, 2010; Hargreaves & Crabtree, 2011). The critical enzymes within the SWI/SNF complexes in human cells are the mutually exclusive proteins Brahma-related gene 1 (BRG1) and Brahma (BRM) (Wang et al, 1996). The mechanisms allowing BRG1 or BRM to bind to chromatin and their interactions with various transcription factors were elucidated in many genetic models (Hoffman et al, 2014; Wilson et al, 2014; Lazar et al, 2020; Orlando et al, 2020). Other conserved protein domains in BRG1 or BRM play functional roles including mediating protein–protein interactions and binding to the acetylated lysines that can be found on histones (Trotter & Archer, 2008).

Although extensive work has been done to characterize the ATPase domain of BRG1, other domains have been demonstrated to also play key roles in BRG1 function (Szerlong et al, 2008; Trotter et al, 2008; Pan et al, 2019). One domain of interest found near the N-terminal region of both BRG1 and BRM is the helicase/SANT-associated (HSA) domain, which has been demonstrated in previous studies to play a critical role in the normal function of BRG1. This domain is predicted to serve as a binding partner for other SWI/SNF proteins such as ARID1A, BAF53a, and actin (Trotter et al, 2008). Within this domain, there are many cancer-associated mutations, and therefore, understanding underlying mechanisms that this domain plays in biology is critical for understanding fundamental mechanisms of chromatin remodeling, SWI/SNF complex formation, and cancer (Sankareswaran et al, 2018).

Recent studies investigating the structure and assembly of SWI/SNF complexes revealed how cancer-associated mutations in SWI/SNF factors could alter BAF complex functions, which result in cancer phenotypes (Ho et al, 2009; Alpsoy & Dykhuizen, 2018). In addition, structural studies demonstrated how the SWI/SNF proteins interact to remodel chromatin, which provides potential links between mutations in complex members and diseases such as cancer (Kadoch et al, 2013; Mashtalir et al, 2018, 2020; Pan et al, 2019; Han et al, 2020; He et al, 2020). Structural studies of the SWI/SNF complexes have further supported the idea of subcomplexes that play unique biological roles based on the SWI/SNF members that make them up. These subcomplexes include the BRG1/BRM-associated factor (BAF), the polybromo-associated factor (PBAF), embryonic-specific BAF (esBAF), and the non-canonical BAF (ncBAF) complexes (Ho et al, 2009; Alpsoy & Dykhuizen, 2018). When examining BRG1 specifically in the study of BAF complex structure and formation, BRG1's HSA domain plays a key role in linking the ATPase module to the core module that interacts with many BAF complex

[1]Chromatin and Gene Expression Section, Epigenetics and Stem Cell Biology Laboratory, National Institute of Environmental Health Sciences, National Institutes of Health, Durham, NC, USA   [2]Integrative Bioinformatics, Epigenetics and Stem Cell Biology Laboratory, National Institute of Environmental Health Sciences, Research Triangle Park, NC, USA

Correspondence: archer1@nih.gov

members (Patel et al, 2019; He et al, 2020; Mashtalir et al, 2020). Analysis of BRG1 structure suggests that the HSA domain interacts with several subunits, including BAF53a/ACTL6A, actin, and the B-cell lymphoma 7 proteins.

The B-cell CLL/lymphoma 7 protein family members A, B, and C are SWI/SNF complex members that bind to BRG1 and are associated with cancer incidence and progression (Kadoch et al, 2013). BCL7A is the most well-studied BCL7 protein because of its role in the pathogenesis of Burkitt's lymphoma as part of a three-gene translocation event (Zani et al, 1996). To date, very little work has been performed investigating BCL7 proteins within the SWI/SNF complex and how they impact nucleosome remodeling. Previous studies suggested that BCL7 proteins bind to BRG1, but those studies have not expanded upon the functions of the BCL7 family members in the complex (Mashtalir et al, 2018). Efforts to characterize the roles in SWI/SNF complexes and their interactions with the various complex members are critical for understanding how individual SWI/SNF proteins can drive normal cellular function and diseases such as cancer.

To understand how the HSA domain functions in BRG1 processes, we generated cell lines that upon chemical induction express WT BRG1 (iBRG1) or BRG1 with the HSA domain deleted (iΔHSA). We identified a significant number of genes that are transcriptionally altered upon iBRG1, but not iΔHSA, expression. The iBRG1-dependent genes drive a signature of senescence and extracellular matrix remodeling, suggesting BRG1 drives a unique gene signature in SW-13 cells that may engage with cancer-related pathways. We also identified novel protein–protein interactions between the HSA domain and the BCL7A/B/C proteins. The interaction between the HSA domain and the BCL7 proteins was critical for the function of BRG1. These data support the model in which the HSA domain of BRG1 helps incorporate the BCL7 proteins for chromatin remodeling and induces transcription in gene expression pathways such as the matrisome and senescence.

# Results

## The HSA domain of BRG1 is necessary to drive a full transcriptional program

As previous efforts have established the requirement of the HSA domain for BRG1 function, we sought to further understand how this domain supports the function of BRG1, especially in light of the recent structural analyses highlighting the critical role the HSA domain plays in linking the BAF complex to the nucleosome (Han et al, 2020; He et al, 2020; Wagner et al, 2020). We used the adrenocortical carcinoma SW-13 cell model in which BRG1 and BRM are not expressed because of epigenetic silencing to examine HSA domain function (Davis et al, 2016). To overcome the reduced proliferation phenotype observed by constitutive BRG1 expression in SW-13 cells (Shanahan et al, 1999), we generated stable SW-13 cell lines that induce the expression of WT BRG1 (iBRG1) or BRG1 with a deletion of the HSA domain (iΔHSA) via treatment with doxycycline. The stable cells selected expressed no detectable levels of iBRG1 or iΔHSA protein under normal conditions and demonstrate a high

level of expression after 24 h of induction by doxycycline (Fig 1A). To characterize the function of the inducible iBRG1 or iΔHSA proteins, we examined the effect of induction of either BRG1 protein on gene expression. Induction of iBRG1 protein increased the expression of typical BRG1 target genes, whereas induction of iΔHSA protein showed no to minor changes in expression (Fig 1B). We tested multiple clonal lines of iΔHSA and demonstrated similar protein expression levels and changes in target gene expression (Fig S1, top). These results indicated that the HSA domain is required for the change in the expression of the characterized BRG1 target genes.

We used RNA sequencing (RNA-seq) to test our prediction that iΔHSA is incapable of driving changes in BRG1-driven gene expression. Analysis of the transcriptomes of iBRG1- or iΔHSA-expressing cells identified 256 differentially expressed genes (DEGs) when iBRG1 was expressed compared with control cells, but only 68 DEGs when iΔHSA was expressed compared with control cells (Fig 1C). These results supported our prediction that the HSA mutant displayed a reduced ability to alter the expression of BRG1 target genes. We supported the observation that several iBRG1 target genes identified in the RNA-seq experiment were either unaffected by iΔHSA expression or showed a significant decrease in the magnitude of expression when compared to iBRG1-expressing cells by quantitative real-time polymerase chain reaction (qRT–PCR) (Fig 1D).

To characterize potential downstream functions because of transcriptional changes from iBRG1 or iΔHSA expression, we used pathway analysis on the DEGs and identified a significant number of pathways implicated in oncogenesis and senescence, including cell cycle, p53 regulation, DNA replication, and DNA damage signal transduction (Fig S1, middle and bottom). Although senescence has been long linked to anti-cancer effects, there is recent evidence that suggests the senescence-associated secretory phenotype (SASP) (Campisi, 1997) can contribute to the tumor microenvironment by increasing inflammation and altering the extracellular matrix (Campisi, 1997; Mavrogonatou et al, 2020). In addition, the activation of p53 would appear to be anti-cancer. However, SW-13 cells express a mutant p53, which contributes to cancer progression and tumor microenvironment support (Sampaoli et al, 2012; Mantovani et al, 2018). We also observed significant enrichment of matrisome and extracellular matrix–associated genes in the iBRG1 set (Fig S1, middle and bottom), suggesting there are BRG1-driven gene expression signatures that are highly implicated in cancer metastasis and the tumor microenvironment (Yuzhalin et al, 2018; Socovich & Naba, 2019). The results demonstrated here support a model in which re-expression of BRG1 in cells that do not express either SWI/SNF ATPase protein drives the expression of genes that would affect the development and progression of cancer, even via pathways that would normally undermine tumorigenesis.

## Long-term BRG1 expression contributes to gene expression profiles related to proliferation and the matrisome

To examine effects on proliferation and growth of extended iBRG1 or iΔHSA re-expression, we cultured cells continuously in vehicle or doxycycline for 14 d, the time at which SW-13 cells expressing BRG1 constitutively display senescence. There were a significant decrease in cellular proliferation when iBRG1 was expressed, and a

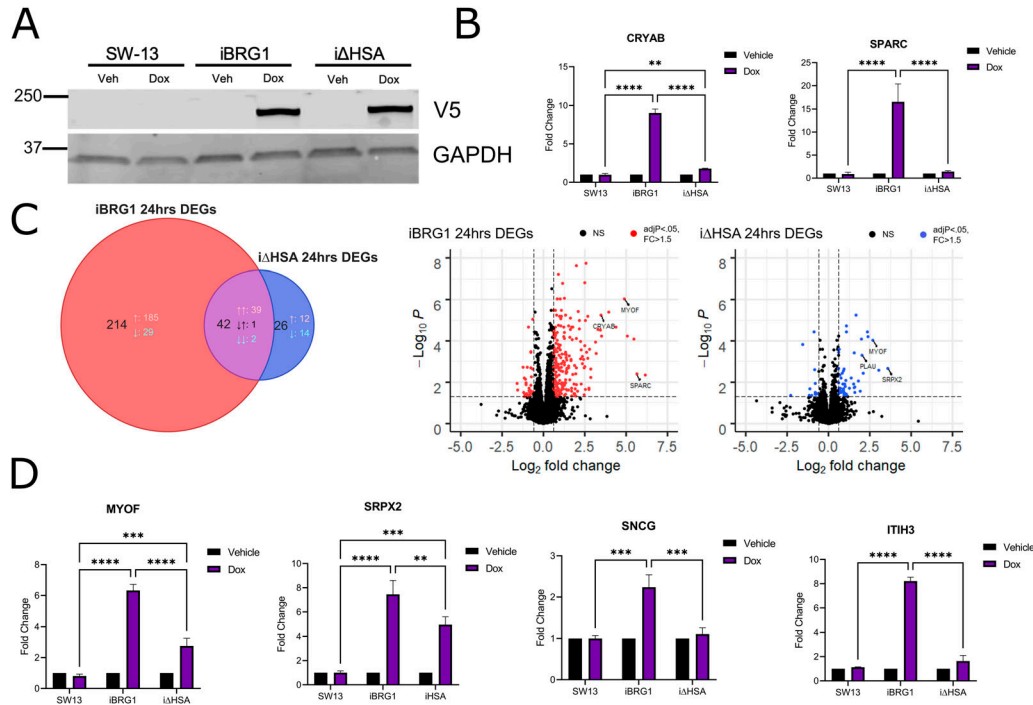

**Figure 1.  HSA domain of BRG1 is necessary to drive a cancer- and senescence-associated gene expression profile.**
**(A)** Detection by Western blot of iBRG1 and iΔHSA proteins in SW-13 cells. Whole-cell lysates were probed with anti-V5 antibody (top) or anti-GAPDH (bottom) from stable cell lines treated with 10 μg/ml doxycycline for 24 h. **(B)** Real-time quantitative PCR (qRT-PCR) was used to determine the gene expression of BRG1 target genes in SW-13, iBRG1, and iΔHSA cells in vehicle- or doxycycline (Dox)-treated conditions. Data are the fold change compared with the vehicle conditions from three biological replicates, and error bars represent the SD. ** represents $P < 0.01$, *** represents $P < 0.001$, and **** represents $P < 0.0001$ (two-way ANOVA). **(C)** Left: Venn diagram of the overlapping numbers of genes comparing iBRG1 (red) and iΔHSA (blue) cells after 24 h of induction by doxycycline. Right: volcano plots displaying the differentially expressed genes identified in iBRG1 (left, red) or iΔHSA (right, blue) cells treated for 24 h of doxycycline compared with vehicle-treated cells. **(D)** Real-time quantitative PCR of newly identified target genes of iBRG1 or iΔHSA after 24 h of doxycycline treatment. Data are the fold change compared with the control conditions from three biological replicates, and error bars represent the SD. ** represents $P < 0.01$, *** represents $P < 0.001$, and **** represents $P < 0.0001$ (two-way ANOVA).

lesser growth phenotype when iΔHSA was expressed instead (Fig S2A). The iBRG1 cells also displayed the SASP much more significantly after 14 d of BRG1 expression, which was not observed in control conditions in either iBRG1 or iΔHSA cells or in induced conditions of iΔHSA cells (Figs S2B and S3). The results observed here suggest that the HSA domain is necessary for the function of BRG1 to drive senescence and reduction in cell proliferation in SW-13 cells.

To identify what gene expression changes were responsible for the growth and senescence phenotypes we observed after continuous iBRG1 expression, we used RNA sequencing to examine the transcriptomes of iBRG1 and iΔHSA cells cultured after the 14-d timepoint in vehicle- and doxycycline-treated conditions. There was a large increase in transcriptional changes in both iBRG1 and iΔHSA cells after 14 d of continuous doxycycline (Fig S2C and D). We detected 2,993 DEGs when considering both iBRG1 and iΔHSA expression (Fig S2C and D). Most of the DEGs were found in the iBRG1 cells (2,622 DEGs; Fig S2C and D) compared with the iΔHSA cells (214 DEGs). There were many more up-regulated DEGs than down-regulated DEGs (2,785 total up-regulated versus 412 total down-regulated genes), suggesting that BRG1 functions mostly to increase the expression of genes in SW-13 cells (Fig S2C and D).

Pathway analysis again demonstrated that many genes altered by iBRG1 expression were significantly enriched in senescence and cancer-associated pathways, whereas a smaller number of pathways were also enriched in genes altered by iΔHSA expression (Fig S2E). Because of the large number of pathways identified in greater than 2,500 DEGs, we focused on the most significant pathways identified, which included the matrisome and extracellular matrix pathways (Fig S2E). We also observed enrichment in other pathways related to oncogenesis and senescence including P53, cytokine signaling, focal adhesion, and PI3K/AKT signaling in iBRG1 cells especially. These data corroborate previous observations of an increase in metastasis signals paired with reduced individual cellular proliferation in SW-13 cells when BRG1 was expressed upon epigenetic activation (Davis et al, 2016). In addition, in cells expressing iΔHSA, the enrichment of pathways was severely reduced, which suggests that this domain plays a critical function in SWI/SNF-driven changes in gene expression.

## iBRG1 and iΔHSA are localized to transcription start sites

One mechanism that could drive the changes in gene expression we observed when comparing iBRG1 and iΔHSA cells is the difference in their ability to bind to chromatin. We used Cleavage Under Targets and Release Under Nuclease (CUT&RUN) sequencing to determine the binding sites of iBRG1 and iΔHSA (Skene & Henikoff, 2017). There was enrichment specifically at the

transcription start sites and gene bodies of genes that are regulated by iBRG1 or iΔHSA expression (SRPX2 and MYOF) (Fig 2A). We determined that the loss of the HSA domain does not impair the ability of BRG1 to correctly localize, as there was significant enrichment of signal but at a comparable level for both iBRG1 and iΔHSA across the genome upon induction (Fig 2B). The enrichment of iBRG1 or iΔHSA was found to generally localize to promoters of annotated genes in SW-13 cells (Fig 2B). Both iBRG1 and iΔHSA also localize to transcription termination sites, but at much lower levels than at the TSS. We then called peaks for the CUT&RUN datasets and determined a similar number of peaks and coverage at the identified peaks by both iBRG1 and iΔHSA (Fig 2C and D). We identified ~11,000 peaks identified as binding sites for iBRG1 and 10,000 peaks identified as binding sites for iΔHSA. The overlap of peaks was similar, and both sets of peaks demonstrated a similar enrichment at genomic sites (Fig 2D and E). These genomic sites include enrichment at promoters and introns/gene bodies when compared to the annotation of the genome (Fig 2E). The analysis of CUT&RUN peaks demonstrated that iBRG1 and iΔHSA bind within or immediately upstream of genes in SW-13 cells and likely participate in the regulation of those genes. These results suggested that iBRG1 can act locally to bind and result in altered gene expression, that the HSA domain is dispensable for proper localization to chromatin and that the HSA domain's role is not for DNA site binding specification.

## BRG1 requires the HSA domain to alter chromatin accessibility

One predicted outcome of chromatin remodeling activity is a change in chromatin accessibility. To test the effect expression of the inducible BRG1 proteins on chromatin accessibility, we used the assay for transposase-accessible chromatin (ATAC) sequencing, comparing iBRG1 and iΔHSA cells, as well as wild-type SW-13 cells, in normal and induced conditions. We identified a similar number of peaks that were highly overlapping in normal conditions in all the cell types, but significantly more peaks in iBRG1 cells after induction (Fig 3A, top left). To identify changes in accessibility, we compared the peaks and the coverage at those sites and called a fold change to identify regions of differentially accessible chromatin (DAC). Upon induction of BRG1 expression, we observed a significant fold change in DACs in iBRG1 cells, a much smaller number of DACs in iΔHSA cells, and near-zero DACs in normal SW-13 cells (Fig 3A, top left). The iΔHSA DACs were mostly found to also be identified as iBRG1 DACs (Fig 3A, top right), which suggests that the deletion of the HSA domain is not resulting in the appearance of differentially accessible sites across the genome, but instead a lack of changes in accessibility. The change in DACs seen mostly by iBRG1 induction, but not iΔHSA induction, agrees with the lack of gene expression changes observed by iΔHSA induction when compared to iBRG1 induction.

We next examined the distribution of peaks across genomic regions and found that the distribution was highly similar across all the sample types; however, in iBRG1-induced samples there was a shift from promoter regions to introns and extragenic regions (Fig 3A, bottom; Fig 3B, top). When examining the annotated chromatin states of the peaks, there was a clear gain in peaks found in enhancer regions in iBRG1-induced samples (Fig 3B, bottom). This

suggests that upon iBRG1, but not iΔHSA, expression, there is an increase in the accessibility of enhancer regions that likely contributes to the gene expression changes.

Next, we examined the ATAC-seq coverage at the different groups of DACs identified, and we plotted the ATAC signal as heatmaps from SW-13, iBRG1, and iΔHSA cells after Dox induction at these sites and scaled the heatmaps to have similar windows, because of having much larger numbers of iBRG1 DACs compared with any other group (Fig 3C). In the SW-13 cells, which do not express either BRG1 or BRM, the SWI/SNF remodeling enzymes, we see limited signal at any of the DAC types identified. When we compare ATAC-seq signal of iBRG1- or iΔHSA-expressing cells at the iBRG1, iΔHSA, or shared iBRG1+ iΔHSA DACs, we observe strong coverage of ATAC signal in iBRG1-expressing cells at iBRG1 and shared DACs, which include the overlapping iΔHSA DACs (Fig 3C). We also see limited ATAC signal in iΔHSA-expressing cells at iBRG1 DACs, but strong ATAC coverage at iΔHSA DACs and shared DACs (Fig 3C). The shared DACs appear to be wider than either the iBRG1- or iΔHSA-specific DACs and show higher signal (Fig 3C, top). This could suggest that the shared DACs are sites where the barrier to change chromatin accessibility is very low, and even with a mutation in the HSA domain of BRG1, remodeling and significant accessibility changes can still occur.

In comparison with the differentially accessible regions, we observed that at all peaks there was relatively similar ATAC-seq signal in iBRG1- or iΔHSA-expressing cells, except for slightly higher signal in iBRG1-expressing cells at iBRG1 peaks compared with iΔHSA-expressing cells at iBRG1 peaks (Fig 3D). This difference in signal can likely be attributed to those novel iBRG1 peaks. When we look directly at the coverage at genes of interest, we can see increases in ATAC signal in iBRG1-expressing cells but limited or no increase in signal in iΔHSA-expressing cells, at genes that show expression changes by iBRG1 expression, but not iΔHSA expression, such as CRYAB, SRPX2, SPARC, and CSF1 (Fig 3E). These results provide a link between the changes in chromatin accessibility we see in iBRG1-expressing cells compared with iΔHSA-expressing cells and the observed changes in gene expression in those induced cells.

When we examined the binding sites of iBRG1 and iΔHSA identified by CUT&RUN compared with ATAC-seq signal, we identified overlaps between the peaks in both iBRG1 and iΔHSA datasets (Fig 3F). ~30% of iBRG1 binding sites were identified as DACs upon iBRG1 induction, and 10% of iΔHSA binding sites were identified as DACs upon iΔHSA induction (Fig 3F). These results suggest that iBRG1 can bind to sites and directly alter chromatin accessibility, but either those sites are not all captured by CUT&RUN or many changes in accessibility are indirect and the consequence of binding at other sites. Examining the ATAC signal in induced conditions at the iBRG1 and iΔHSA binding sites demonstrated an enrichment of iBRG1 ATAC signal at these sites, but not in other conditions (Fig 3F). This suggests that iBRG1 allows for changing accessibility, likely through remodeling activity, at these sites, but not iΔHSA. The reduced overlap between iΔHSA binding sites and iΔHSA DACs again highlights the ability of iΔHSA to bind to chromatin, but iΔHSA is highly restricted in the ability to alter accessibility, likely though restricted remodeling activity, leading to severely reduced changes in gene expression. The iΔHSA protein

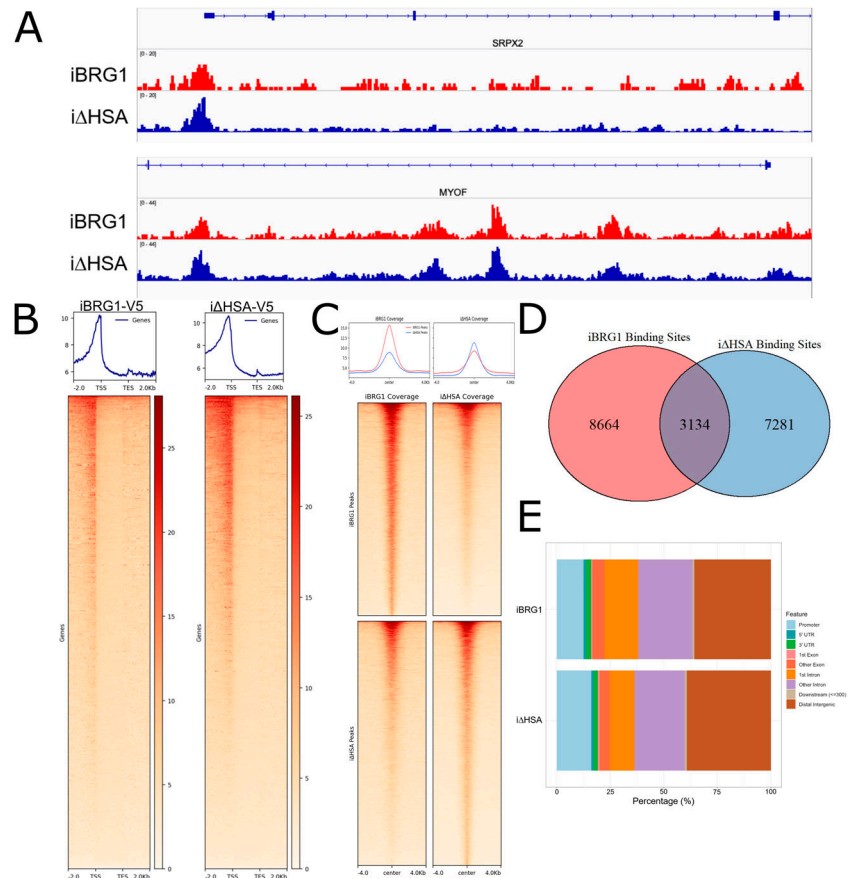

**Figure 2. Loss of the HSA domain does not alter binding of BRG1 to transcriptional start sites.**
**(A)** CUT&RUN coverage of iBRG1 (red) or iΔHSA (blue) at SRPX2 and MYOF, two genes regulated by iBRG1 expression after 24 h. **(B)** Heatmaps and metaplots of iBRG1 or iΔHSA CUT&RUN signal in a 6-kb window around all transcriptional start sites, and they are scaled to represent all gene bodies. **(C)** Heatmaps and metaplots of iBRG1 or iΔHSA CUT&RUN signal at called iBRG1 or iΔHSA CUT&RUN peaks. **(D)** Venn diagram of overlaps between iBRG1 and iΔHSA CUT&RUN peaks. **(E)** Annotation of genomic regions of iBRG1 and iΔHSA CUT&RUN peaks.

appears to allow for ATPase function, as there are some changes in chromatin accessibility, but the explanation for the gene expression phenotype we observe upon iΔHSA expression likely is driven by a different mechanism.

### The HSA domain is necessary for critical interactions between BRG1 and multiple SWI/SNF proteins

We hypothesized that a primary driver of the inability of iΔHSA expression to affect gene expression changes and alter chromatin accessibility to the degree of iBRG1 expression is the failure of iΔHSA to interact with other SWI/SNF complex members. We previously demonstrated that ARID1A was a novel interacting partner with the HSA domain via immunoprecipitation (Trotter et al, 2008) and sought to characterize the full SWI/SNF interactome with iBRG1 or iΔHSA in SW-13 cells. We used co-immunoprecipitation followed by mass spectrometry (IP-MS) to identify the proteins that interact with iBRG1, which are subsequently lost upon HSA domain deletion. We found several interactions that were lost when the HSA domain was deleted from BRG1 (Fig 4A), which included interactions with BAF53a, BCL7A, BCL7B, and BCL7C, notably proteins that have recently been suggested to bind to BRG1 via the HSA domain (Fig 4A) (Marcum et al, 2020). IP-MS also determined interactions between BAF250a and iΔHSA, which were previously undetected by Western

blot, suggesting that this protein does not absolutely require the HSA domain to form a complex with BRG1 (Fig 4A).

To confirm the interactions identified by IP-MS, we repeated the IP experiments and examined the interactions between the HSA-requiring SWI/SNF subunits by Western blot. We observed that BCL7C was capable of immunoprecipitating iBRG1, but not iΔHSA (Fig 4B and C), the reciprocal of the IP-MS experiment. We also detected a failure to pull down BAF53a by iΔHSA, which has been previously demonstrated (Fig 4B). We determined that both iBRG1 and iΔHSA were able to pull down other SWI/SNF complex members such as BAF155 and that iBRG1 and iΔHSA were precipitated by BAF155, so the loss of the HSA domain does not alter all BRG1 to SWI/SNF protein–protein interactions (Fig 4D).

As IP-MS had indicated a failure to pull down BAF45b and GLTSCR1L by iΔHSA, we tested this by Western blot and observed that iΔHSA can interact with GLTSCR1L, but the BAF45b reagent was insufficient to demonstrate any form of pulldown (Fig 4D and E). This suggests that the failure to detect the interaction with many SWI/SNF complex members via mass spectrometry could be an issue of limited concentration and not a failure to interact. These observations lend support to a model in which the HSA domain acts to maintain critical interactions with the SWI/SNF complex, specifically the BCL7 proteins and BAF53a. We predict that these interactions are critical for downstream chromatin remodeling activity, which should be tested in future studies.

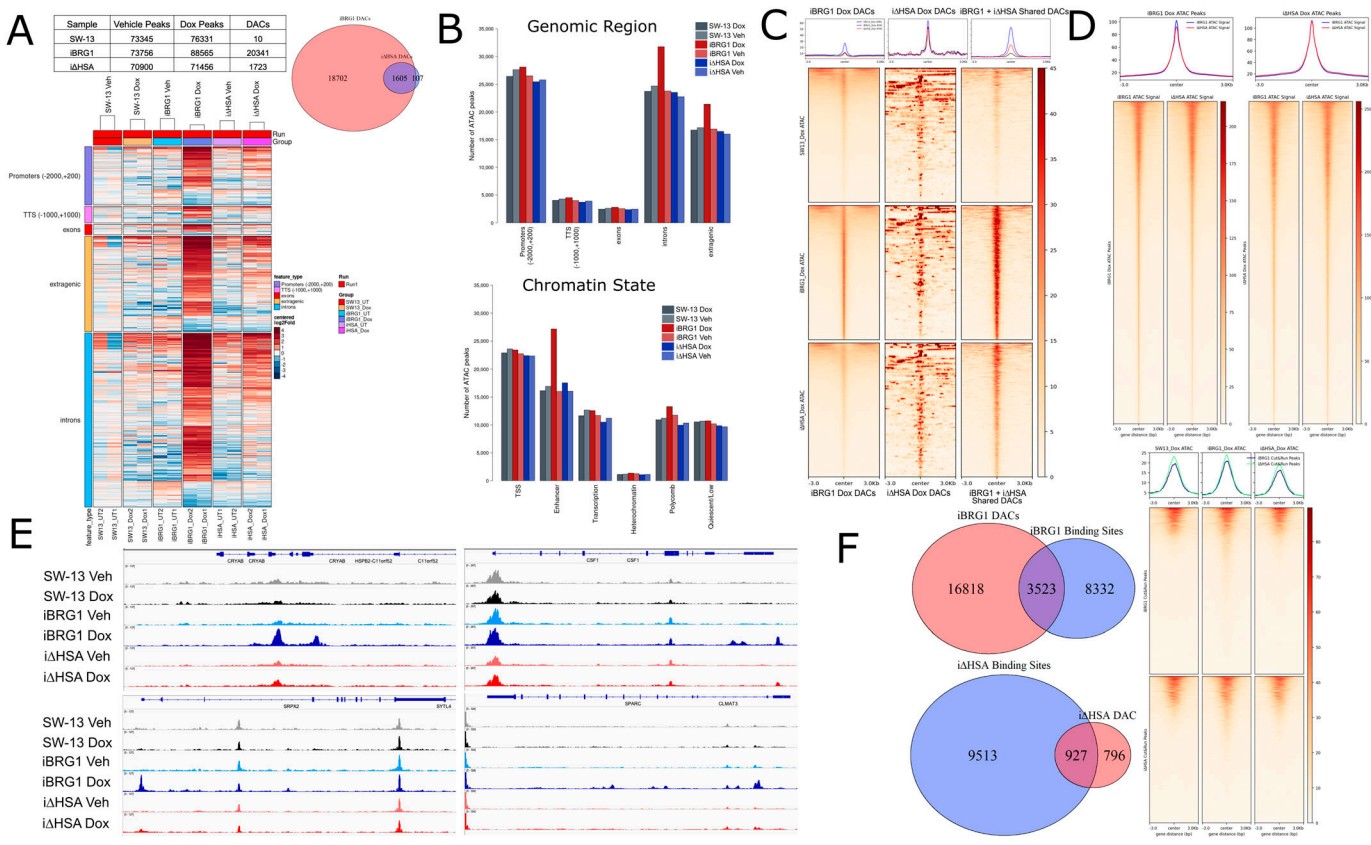

**Figure 3. BRG1 requires the HSA domain to alter chromatin accessibility.**
**(A)** Top left: table of peaks and DACs identified from ATAC-seq in SW-13, iBRG1, or iΔHSA cells. Top right: Venn diagram of DACs identified in iBRG1 or iΔHSA cells comparing Dox with vehicle conditions. Bottom: heatmap of DACs identified by ATAC-seq and the genomic locations in SW-13, iBRG1, or iΔHSA cells. **(B)** Top: percent distributions of each sample peak at genomic location types. Bottom: percent distribution of each sample peaks at annotated chromatin states. **(C)** Top: metaplot profile of ATAC-seq coverage from SW-13, iBRG1, or iΔHSA cells after doxycycline treatment at DACs identified from iBRG1-expressing cells alone and iΔHSA-expressing cells alone, or shared between iBRG1- and iΔHSA-expressing cells. Bottom: heatmap of ATAC-seq coverage at DACs described in the metaplot, scaled so that each window is the same physical size independent of the number of rows. **(D)** Metaplot and heatmap of iBRG1 or iΔHSA at all peaks from iBRG1-expressing cells or iΔHSA-expressing cells. **(E)** ATAC-seq coverage of all samples at CRYAB, CSF1, SRPX2, or SPARC. **(F)** Left top: Venn diagrams of iBRG1 Dox ATAC-seq peaks and iBRG1 CUT&RUN peaks. Left bottom: Venn diagrams of iΔHSA Dox ATAC-seq peaks and iΔHSA CUT&RUN peaks. Right: heatmaps and metaplots of SW-13, iBRG1, or iΔHSA ATAC-seq coverage at iBRG1 or iΔHSA CUT&RUN peaks.

### The HSA domain of BRG1 is necessary for the interaction of HSA domain-interacting proteins with chromatin

To characterize how the interaction with BCL7 proteins could be critical for BRG1 function, we examined the protein and mRNA expression levels of BCL7A, BCL7B, and BCL7C when iBRG1 or iΔHSA was expressed in SW-13 cells. There were no detectable changes in the mRNA of BCL7A, BCL7B, or BCL7C by qRT–PCR when comparing iBRG1 or iΔHSA cells after 24 h of vehicle or induced conditions (Fig S4). However, we did observe a change in the protein concentration of BCL7A, BCL7B, and BCL7C (BCL7s) when iBRG1 was expressed in SW-13 cells, which was not observed upon iΔHSA expression (Fig 4F). Our results suggest that the HSA domain of BRG1 is critical for binding to at least one BCL7 protein, is important for increasing or stabilizing the protein concentrations of all three BCL7s, and leads to the prediction that the lack of the HSA domain restricts their chromatin binding, because of their lack of interaction with a critical member of the SWI/SNF complex.

To test the association of SWI/SNF remodeling complex members, including the BCL7s, with chromatin, we used salt fractionation. We observed a similar affinity for chromatin by iBRG1 and iΔHSA, suggesting the HSA domain is not necessary for chromatin binding (Fig 4G). Just as iBRG1 increased the protein concentration of BCL7C in cells, iBRG1 expression also resulted in a change in the elution pattern of BCL7C (Fig 4H). There was little to no detection of BCL7C expression or elution in iΔHSA-expressing cells, which could suggest a few potential mechanisms for this observation. One such mechanism could be that the HSA domain increases general complex stability and allows for BCL7C to be retained at chromatin. Alternatively, the domain could directly affect the affinity of BCL7C with chromatin; however, we did not directly test and support either of these potential mechanisms. We were unable to highly detect BCL7A or BCL7B in this assay, but we predict a similar phenotype would be observed for these proteins as with BCL7C.

We then observed that the expression of iBRG1, but not iΔHSA, resulted in BAF53a elution at higher salt concentrations (Fig 4I). We did not observe different chromatin binding when iBRG1 or iΔHSA is

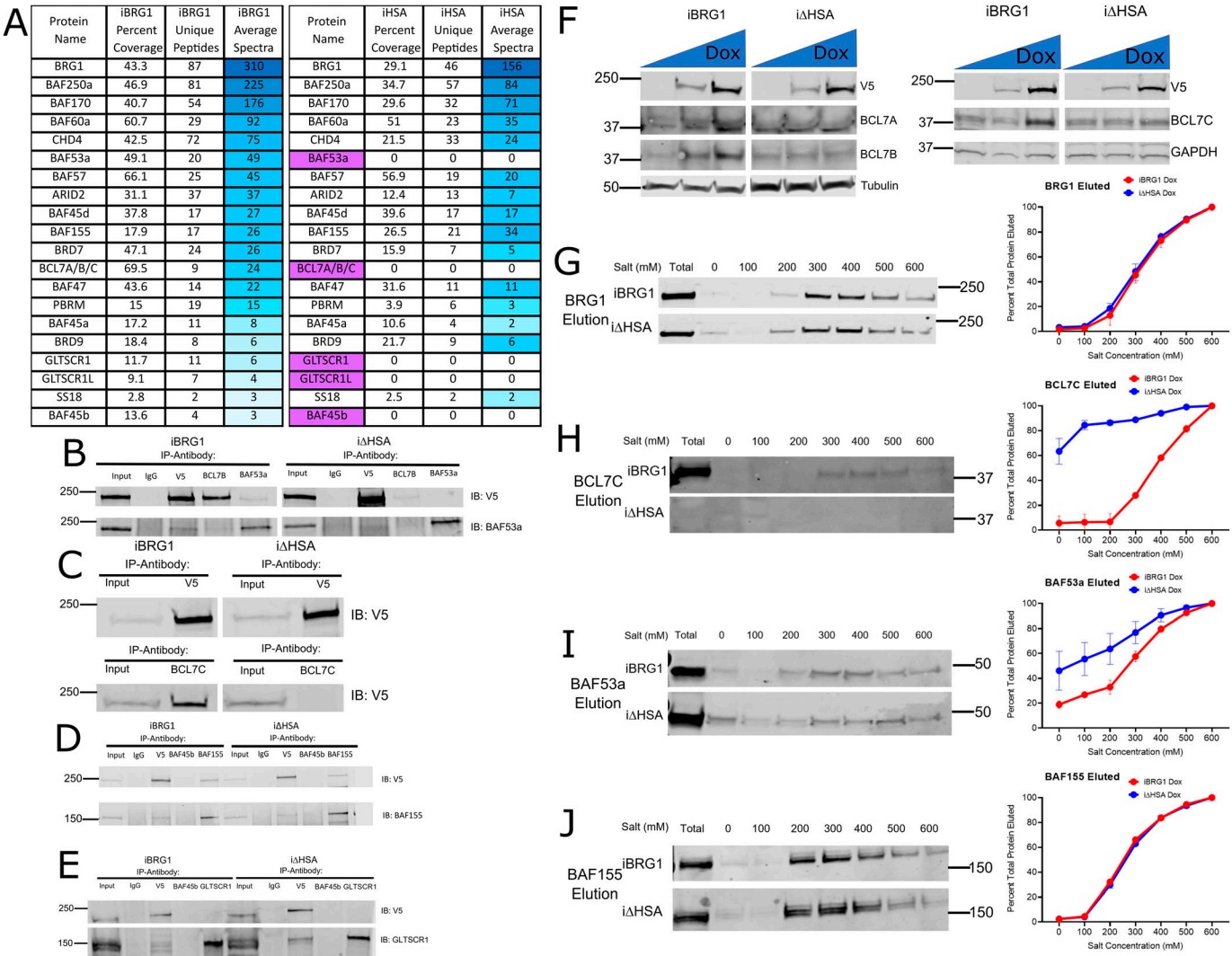

**Figure 4. HSA mutant interacts with a subset of the SWI/SNF complex members bound by WT BRG1 and is necessary for increased BCL7 protein levels.**
**(A)** Co-immunoprecipitation–mass spectrometry data comparing the SWI/SNF complex members identified between the iBRG1 line and the iΔHSA cell line. The SWI/SNF complex protein name is found in the first column, followed by the percent of the protein covered by the mass spectrometry experiment, the number of unique peptides identified and the MS score, and the average spectral counts from each protein pulled down by the V5 antibody used to detect iBRG1 or iΔHSA in the second column. Each column represents the average from three biological replicates. The proteins are colored from blue to white by spectrum average with white being the lowest (zero) and blue being the highest. The protein names highlighted in purple and crossed out were not found in the iΔHSA pulldown. **(B)** Co-immunoprecipitation of SWI/SNF proteins from iBRG1 or iΔHSA cells after 24 h of doxycycline treatment. The top rows indicate the 5% input, or the antibody used for the immunoprecipitation (IgG, V5, BCL7B, or BAF53a), and the labels on the right indicate the antibody used for detection on the immunoblot (top: V5; bottom: BAF53a). **(C)** Co-immunoprecipitation displaying the interaction between BCL7C and BRG1 in iBRG1 or iΔHSA cells after 24 h of doxycycline treatment. The top rows indicate the 5% input, or the antibody used for the immunoprecipitation (top: V5; bottom: BCL7C), and the label on the right indicates the antibody used for detection on the immunoblot (V5). **(D)** Co-immunoprecipitation displaying the interaction between BAF155 and iBRG1 or iΔHSA. The top rows indicate the 5% input, or the antibody used for the immunoprecipitation (IgG, V5, BAF45b, or BAF155), and the label on the right indicates the antibody used for the detection on the immunoblot (V5 or BAF155). **(E)** Co-immunoprecipitation displaying interaction between GLTSCR1L and BAF45b with iBRG1 or iΔHSA. The top rows indicate the 5% input, or the antibody used for the immunoprecipitation (IgG, V5, BAF45b, or GLTSCR1L), and the label on the right indicates the antibody used for the detection on the immunoblot (V5 or GLTSCR1L). **(F)** Immunoblot detection of BRG1, BCL7A, BCL7B, and tubulin (left) or BRG1, BCL7C, and GAPDH (right) protein levels with increasing levels of iBRG1 or iΔHSA expression by doxycycline treatment. Each blot contains equivalent levels of protein per lane, and the levels of doxycycline used for 24 h increase from left to right. **(G)** Differential salt extraction of BRG1 in iBRG1 or iΔHSA cells after 24 h of doxycycline treatment. Left is the immunodetection of BRG1 or iΔHSA in doxycycline-treated conditions. Right is the average fraction of the total protein extracted in each salt concentration in iBRG1 or iΔHSA cells from two replicates. Error bars represent the SD. **(H)** Differential salt extraction of BCL7C in iBRG1 or iΔHSA cells after 24 h of doxycycline treatment. Left is the immunodetection of BRG1 or iΔHSA in doxycycline-treated conditions. Right is the average fraction of the total protein extracted in each salt concentration in iBRG1 or iΔHSA cells from two replicates. Error bars represent the SD. **(I)** Differential salt extraction of BAF53a in iBRG1 or iΔHSA cells after 24 h of doxycycline treatment. Left is the immunodetection of BRG1 or iΔHSA in doxycycline-treated conditions. Right is the average fraction of the total protein extracted in each salt concentration in iBRG1 or iΔHSA cells. Error bars represent the SD. **(J)** Differential salt extraction of BAF155 in iBRG1 or iΔHSA cells after 24 h of doxycycline treatment. Left is the immunodetection of BRG1 or iΔHSA in doxycycline-treated conditions. Right is the average fraction of the total protein extracted in each salt concentration in iBRG1 or iΔHSA cells. Error bars represent the SD.
Source data are available for this figure.

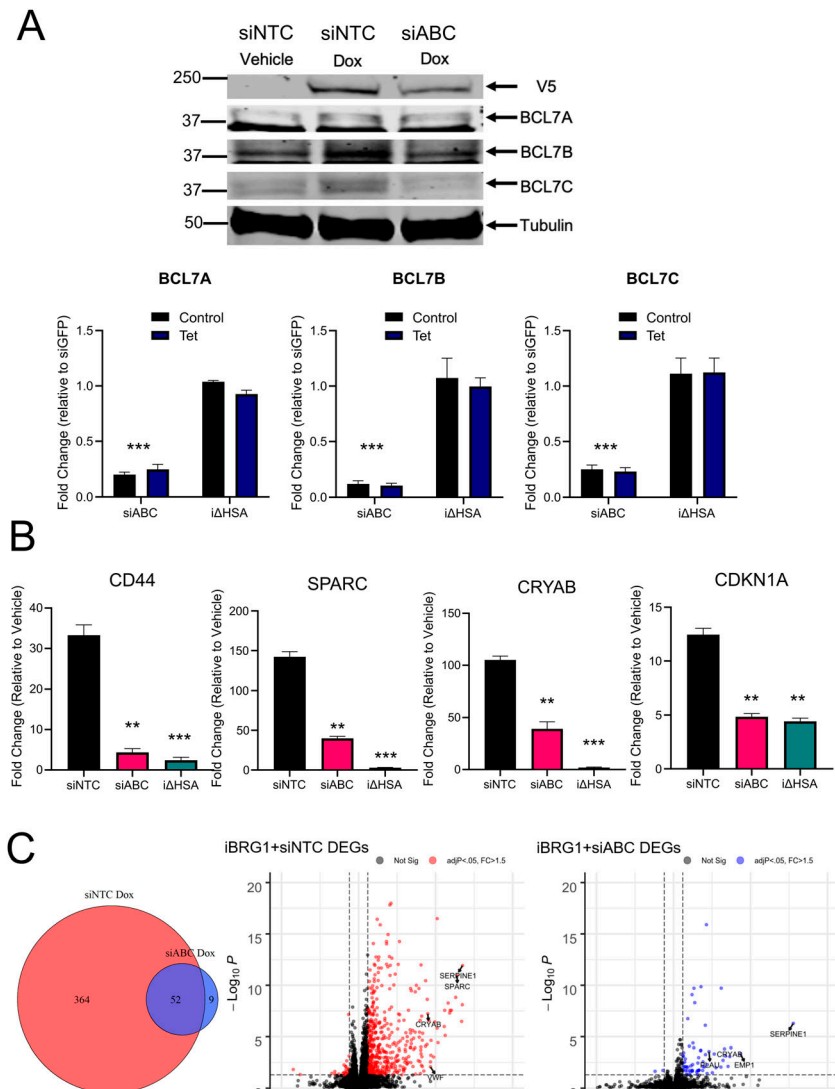

**Figure 5.  BCL7 proteins are necessary for gene expression changes driven by BRG1.**
**(A)** Top: immunoblot detection of BRG1-V5, tubulin, BCL7A, BCL7B, or BCL7C in iBRG1 cells with and without knockdown of BCL7A, BCL7B, and BCL7C by siRNA. Left is vehicle-treated iBRG1 cells, center left is iBRG1 cells after 24 h of doxycycline treatment, center is iBRG1 cells after 24 h of doxycycline treatment, and right is iBRG1 cells after 24 h of doxycycline treatment and BCL7A, BCL7B, and BCL7C combined knockdown. Bottom: real-time quantitative PCR of BCL7A, BCL7B, or BCL7C expression in iBRG1 cells after knockdown of BCL7A, BCL7B, and BCL7C or in iΔHSA cells with or without doxycycline treatment. **(B)** Real-time quantitative PCR of BRG1 target genes (CD44, CDKN1A, CRYAB, or SPARC) after knockdown in iBRG1 cells of GFP (negative control) or BCL7A + BCL7B + BCL7C followed by 24 h of doxycycline treatment. Data are the fold change compared with the control conditions in each siRNA or cell line condition from three biological replicates, and error bars represent the SD. * represents $P < 0.05$, ** represents $P < 0.01$, and *** represents $P < 0.001$, when compared to siNTC conditions (Welch's $t$ test). **(C)** Left: Venn diagram of DEGs identified by RNA-seq after knockdown BCL7A + BCL7B + BCL7C or a non-template control in iBRG1 cells after 24 h of induction by doxycycline. Right: volcano plots of DEGs identified by RNA-seq in iBRG1 cells after knockdown of a non-template control (left, red) or of BCL7A + BCL7B + BCL7C (right, blue).

for the SWI/SNF complex member BAF155 (Fig 4J), a protein that does not require the HSA domain to interact with BRG1. These results support a model in which the BCL7 proteins are lowly expressed or unable to interact with the SWI/SNF complex without BRG1. Upon BRG1 expression, the interaction with the BCL7 proteins allows for the formation of a complex that can stably interact with chromatin to participate in altering chromatin accessibility with the other members of the SWI/SNF complex. The absence of an HSA domain within BRG1 disrupts this stabilization and binding, leading to the difference in gene expression and accessibility that was observed.

## The BCL7 proteins are required for changes in gene expression driven by BRG1

We predicted that the interaction with the BCL7 proteins is critical for the function of iBRG1 in SW-13 cells, and the loss of the interaction between iBRG1 and BCL7 proteins drives the phenotypes observed when iΔHSA is expressed. To characterize this functionality, we used siRNAs targeting the BCL7s as a group (siABC) in combination with the inducible expression system of BRG1 in SW-13 cells to monitor changes in gene expression. We observed strong loss of BCL7C protein, medium knockdown of BCL7B protein, and minimal, if any, loss of BCL7A protein in SW-13 cells when transfected with siRNAs against them (Fig 5A, top). When we then examined the siRNA effect on the RNA expression of the BCL7 genes, we observed strong knockdown of all BCL7 mRNAs by siRNA transfection (Fig 5A, bottom; Fig S5). This suggests that the antibody reagent against BCL7A is likely not specific enough to detect any potential loss of protein. After knockdown, we observed a reduction in the induction of many target genes when the BCL7s were knocked down by siRNAs against the BCL7s (Fig 5B). This reduced expression of target genes after BCL7 knockdown suggested that the BCL7s are important for BRG1 transcriptional activation. To further test the

requirement for the BCL7 proteins to function in BRG1-driven transcription, we tested the changes in gene expression using polyA-selected mRNA sequencing after temporary knockdown of the BCL7 proteins compared with a non-targeted control (Fig 5C). We observed a slightly higher number of DEGs in iBRG1 cells when examining induction by doxycycline- versus vehicle-treated conditions compared with our first RNA-seq but with a high degree of overlap. This difference is likely seen in the use of polyA selection versus total RNA, sequencing depth, and potentially anti-sense transcription detection differences (Chao et al, 2019). Like our observations of iΔHSA-driven transcription, knockdown of BCL7A, BCL7B, and BCL7C reduced the changes in gene expression driven by BRG1, as only 61 BRG1-dependent DEGs were identified in siABC cells (Fig 5C, left). We again observed a bias toward up-regulation of genes in both the non-template control knockdown cells and the BCL7 knockdown cells, reinforcing the model that BRG1 primarily functions to up-regulate genes and that the HSA domain of BRG1 and its interaction with the BCL7s are important for appropriate function (Fig 5C, middle and right). Overall, the phenotype observed by the loss of the HSA domain was found to be highly similar to the knockdown of the BCL7 proteins, supporting the hypothesis that BCL7 proteins are key for the function of BRG1 to drive transcriptional changes. The results reinforce the model that the HSA domain is essential for the interaction of BRG1 with the BCL7 proteins, and this interaction is required for BRG1-driven SWI/SNF function.

## Discussion

The relationship between SWI/SNF, BRG1, and many diseases including cancer has been described in detail, but the specific mechanisms that are responsible for those interactions continue to be discovered (Pan et al, 2019; Schick et al, 2019; Sobczak et al, 2020). One limitation of previous studies of BRG1-specific function has been the presence of remaining ATPase activity by BRM, which could potentially rescue any functional loss of BRG1. This study begins to address these limitations by using a cell system that lacks both BRG1 and BRM and provides insights into the immediate- and longer term consequences of re-introducing an individual chromatin remodeler into a cancer cell type.

The effects of the re-introduction of BRG1 are most clear in the transcriptional changes that occur, with a significant number of pathways that are often associated with cancer becoming activated even 24 h after expression of a WT BRG1 protein. The significant difference in the transcriptional profiles of the cells expressing a BRG1 protein lacking the HSA domain compared with cells expressing WT BRG1 demonstrates the high level of importance this domain has for BRG1 function. We demonstrated that this domain is not necessary for localization to chromatin, which suggests an effector binding partner that drives this difference.

As we have previously demonstrated that BRG1-ΔHSA is capable of remodeling nucleosomes in an in vitro system (Trotter et al, 2008), the phenotypes observed in this study suggest that the process of SWI/SNF complex formation and subsequent remodeling action may be specifically altered by the loss of this domain. However, the loss of the HSA domain did not appreciably affect the

localization of BRG1 to promoters. We subsequently examined the alteration of chromatin accessibility by ATAC-seq in vehicle- and doxycycline-induced conditions in SW-13, iBRG1, and iΔHSA cells, and observed a significant change in accessibility upon iBRG1 induction and a severely reduced change in iΔHSA cells. The binding sites that we identified by CUT&RUN were also found to be sites where chromatin accessibility was altered by iBRG1 expression, and less so by iΔHSA expression. The genomic and chromatin regions that iBRG1 altered accessibility were strongly enriched at enhancer sites, which suggests a mechanism in which iBRG1, but not iΔHSA, does not predominantly bind to promoter-proximal regions to drive gene expression, but instead at enhancers, to result in alterations in gene expression. We did observe that iΔHSA was able to remodel significantly fewer sites, which supports our previous data that the deletion of the HSA domain does not inhibit the direct enzymatic function of BRG1, but likely interrupts other critical functions of the protein, including the interaction of BRG1 with other components of the remodeling complex to drive complex functionality.

The observation that the loss of the HSA domain, which is required for BCL7 binding to BRG1, and the reduction in BCL7 function have highly similar phenotypes suggests that the BCL7 proteins play an essential role in BRG1 and SWI/SNF function. The BCL7 family of proteins were discovered by their relationship to the B-cell lymphomas that provide their names. The original discovery of BCL7A in a Burkitt lymphoma cell line suggested a potential interaction with actin, which would be a novel unexplored SWI/SNF mechanism as the BCL7 proteins bind to the HSA domain, which was previously described as an actin-related binding protein domain (Zani et al, 1996; Szerlong et al, 2008). Recent work has furthered the understanding of how BCL7A specifically plays a tumor-suppressor role in diffuse large B-cell lymphoma (DLBCL) and demonstrated some preliminary mechanisms of SWI/SNF function in B-cell biology (Baliñas-Gavira et al, 2020). Much less has been described in relation to BCL7B and BCL7C function in normal or cancer biology. BCL7B has been shown to regulate the Wnt signaling pathway, which may drive the relationship of this gene with the genetic disorder, Williams syndrome (Uehara et al, 2015). BCL7C was recently demonstrated to play tumor-suppressive role by interacting with p53 in ovarian carcinomas (Huang et al, 2021). We also observed the requirement of an HSA domain to drive stable protein expression of the BCL7 proteins, which suggests that complex integrity is an essential part of SWI/SNF function. Future studies to examine the establishment of subcomplexes in cells missing individual subunits and their subsequent genomic functions will demonstrate mechanisms that specific complexes play in driving diseases, as they may have very targeted binding sites beyond promoters and altered gene expression profiles in certain cell types.

The long-term expression of iBRG1 in SW-13 cells resulted in extensive gene expression, senescence, and growth changes when compared to the expression of iΔHSA. The changes observed in this study are similar to what has been previously reported in other cell types for re-expression of BRG1 (Lazar et al, 2020; Orlando et al, 2020). The critical differences in this study include a bias toward up-regulated genes and differential pathways in SW-13 cells compared with the other cell types such as A549 and SCCOHT. Notably, one of the key pathways observed in short-term and even more enriched in long-term iBRG1 expression was the matrisome and extracellular matrix gene pathways. We observed six of the nine core

matrisome genes that have been implicated in breast, esophageal, gastric, lung, ovarian, and colorectal cancers to be up-regulated by iBRG1, but not iΔHSA (Yuzhalin et al, 2018). A similar gene signature list of ECM-related prognostic and predictive indicators was highly enriched in iBRG1- but not iΔHSA-expressing cells (Lim et al, 2017). The matrisome genes support primary tumor growth, alter tumor invasiveness, increase angiogenesis, and regulate metastasis (Socovich & Naba, 2019). These pathways were the most significant enriched in our analysis and suggest that BRG1 expression in SW-13 cells drives potential cancer signaling pathways. One such pathway is the induction of senescence that was seen after long-term iBRG1 expression, which can cause inflammation and alter the microenvironment to allow for tumor cell escape (Liu et al, 2018). There have been many recent examples of senescence providing a mechanism for cancer cells to metastasize and increase tumor progression as opposed to fighting cancer (Campisi, 1997; Davis et al, 2016; Mavrogonatou et al, 2020; Ou et al, 2020). In addition, the matrisome and senescence pathways are highly linked in cancer because of the process of inflammation, immune alterations, and epithelial-to-mesenchymal transition (Mavrogonatou et al, 2020). The gene expression signatures here support potential disease models of BRG1 contributing significantly to cancer progression in multiple avenues and also provide support for the observations that BRG1 is involved in highly aggressive forms of prostate and skin cancers (Saladi et al, 2010; Muthuswami et al, 2019).

In summary, we have demonstrated that the HSA domain of BRG1 is essential for the function of this chromatin remodeler to drive gene expression signatures that support multiple cancer mechanisms including oncogenic senescence, metastasis, and tumor microenvironment support. A limitation of our study is that we have not biochemically examined remodeling function, but examined the changes in accessibility driven by either iBRG1 or iΔHSA expression by ATAC-seq. In addition, we cannot completely exclude how a shortened BRG1 protein could drive some of the phenotypes observed here. We then found that the HSA domain directly interacts with the understudied BCL7 family of proteins, which are required for the genomic responses we observed by BRG1 re-expression in SW-13 cells. The observations seen here support a mechanism where the BCL7 proteins interact via the HSA domain of BRG1 with the SWI/SNF remodeling complex to drive gene expression changes that result in cancer progression. However, an additional limitation of this study is that we have not directly addressed the mechanism and function of each specific BCL7 protein but focused on the knockdown of all three proteins. In addition, we have not directly characterized how the loss of these proteins alters chromatin remodeling or accessibility. These results provide a pathway for understanding how specific interactions of individual remodeling complex subunits are essential for the function of the entire complex.

# Materials and Methods

### Cell culture

Human SW-13 adrenal carcinoma cells and the iBRG1 and iΔHSA derivatives were maintained as previously described (Leibovitz et al, 1973; Trotter & Archer, 2004). Transfections of siRNAs were carried out in serum-free OptiMEM using Lipofectamine 2000 (Invitrogen) according to the manufacturer's instructions. For short-term induction of iBRG1 or iΔHSA cells, a 1,000X solution of 10 mg per ml of doxycycline hyclate (D9891; Sigma-Aldrich) was diluted to 1X in Dulbecco's minimal essential medium (DMEM).

### Generation of iBRG1 and iΔHSA cells

The coding sequence of full-length *Brg1* or *Brg1* with the HSA domain deleted was amplified by PCR to contain a C-terminal V5-tag and directional SfiI restriction sites from previously generated constructs (Trotter et al, 2008). The PCR products along with the plasmid pSBtet-Neo (#60509; ADDGENE) were digested with SfiI overnight at 37°C, purified by agarose gel electrophoresis, and ligated using T4 ligase (New England Biolabs) (Kowarz et al, 2015). Constructs were transfected into One Shot Top10 competent cells (Thermo Fisher Scientific), and individual colonies were selected and screened for inserts by PCR and Sanger sequencing. 200,000 SW-13 cells were plated in six-well culture plates, and the next day, 1.9 μg of SBtet-iBRG1 or SBtet-iΔHSA was transfected along with 100 ng of the transposase SB100X (#34879; ADDGENE) into the cells. 24 h after transfection, cells were subjected to 250 μg/ml G418 (Thermo Fisher Scientific) and selection was carried out for 5 d followed by individual cell selection by limiting dilution into 96-well plates. Individual colonies were grown and expanded and then tested for inducible BRG1 or HSA expression by Western blot. Individual clones were selected and tested to monitor changes in protein expression levels and changes in gene expression. Multiple clones were compared and demonstrated to have similar changes in target gene expression and protein levels, and individual clones of iBRG1 and iΔHSA were used for the subsequent data collection.

### RNA isolation, cDNA synthesis, qRT-PCR, and RNA sequencing

RNA was isolated from iBRG1 or iΔHSA cells using QIAGEN RNeasy kits with on-column DNase treatment (QIAGEN). cDNAs for individual gene expression–level detection were generated using 1,000 μg of total RNA with the iScript cDNA Synthesis Kit (Bio-Rad). Quantitative real-time PCR was performed using ssoAdvanced Universal SYBR Green Supermix (Bio-Rad). All quantitative real-time PCR was performed on three biological replicates. RNA-sequencing libraries for 24-h Dox induction and long-term Dox induction were generated using ribosomal RNA–depleted libraries using the Ribo-Zero Gold and then the TruSeq Stranded Total RNA library kit (Illumina). mRNA-sequencing libraries for siRNA experiments were generated using polyA-purified RNA and then the Illumina TruSeq RNA Sample Prep Kit v2. The libraries were sequenced using an Illumina NextSeq 500 using 2 × 75 base pair reads for a targeted sequencing depth of 50 million reads per sample for total RNA and 25 million reads per sample for polyA-selected RNA. RNA sequencing was performed for three biological replicates for all cell lines and all treatment conditions. The following primers were used in this study: *Brg1* Forward: AGGACAACATGCACCAGATG, *Brg1* Reverse: CTGACCGCATCCCCATTC; *cryab* Forward: TGTTGGGAGATGTGATTGAGG, *cryab* Reverse: GGGATGAAGTAATGGTGAGAGG; *sparc* Forward: CGA CTCTTCCTGCCACTTC, *sparc* Reverse: GGAATTCGGTCAGCTCAGAG; *csf1* Forward: CGCTTCAGAGATAACACCCC, *csf1* Reverse: TCATAGAAAG

TTCGGACGCAG; *myof* Forward: GGAACTGGGGCTGCATCAT, *myof* Reverse: ACTCCTTCCCCCTTTCCAGT; *srpx2* Forward: TTCAAGGATGGCCAGTCAGC, *srpx2* Reverse: TAACACCATCGGGGGACTCG; *sncg* Forward: ACACTGTGGCCACCAAGAC, *srpx2* Reverse: ACTCTGGGCCTCCTCTGC; *itih3* Forward: GACAACGAGGATGAGAGGGC, *itih3* Reverse: GGGATCCCCGTCCACATAGT; *cd44* Forward: CGTGGAGAAAAATGGTCGCTACAG, *cd44* Reverse: GTGGGCAAGGTGCTATTGAAAGC; *cdkn1a* Forward: CATGGGTTCTGACGGACAT, *cdkn1a* Reverse: AGTCAGTTCCTTGTGGAGCC; *bcl7a* Forward: GGTGACCGTTGGTGACACAT, *bcl7a* Reverse: GGTCACCTCTGAGCCACACT; *bcl7b* Forward: CAGCCCGAGAACCTAATGG, *bcl7b* Reverse: AGACACGGAACTCTGGTTGC; *bcl7c* Forward: CCAGAAGGGTCCCTGAG, *bcl7c* Reverse: GAACAGGCTCCTCCTTGGTC; *tubulin* Forward: GAAGCCACAGGTGGCAAATA, *tubulin* Reverse: CGTACCACATCCAGGACAGA; and *gapdh* Forward: ACAACTTTGGTATCGTGGAAGG, *gapdh* Reverse: GCCATCACGCCACAGTTTC.

## Western blot

To detect iBRG1 or iΔHSA protein levels, cells were lysed in high-salt Buffer X (100 mM Tris–HCl, pH 8, 420 mM NaCl, 1% NP-40, and 1 mM EDTA) containing cOmplete mini protease inhibitors (Roche) for 20 min on ice followed by 5 s of sonication by a probe sonicator. Samples were spun at 16,000$g$ for 10 min at 4°C, and the supernatant was taken as total protein lysate. Protein concentration was determined using the Bio-Rad Protein Assay, and 50 $\mu$g of total protein was loaded onto a 4–20% Criterion Tris–HCl gel and run at 100 V for 90 min. The gels were transferred to nitrocellulose paper (Bio-Rad) for 2 h at 400 mA at 4°C. The blots were blocked for 1 h at room temperature in TBS containing 5% milk. The membranes were incubated overnight at 4°C with the following primary antibodies: anti-V5 (#46-0705; Invitrogen) and GAPDH (DSHB, DSHB-hGAPDH-2G7). The membranes were then washed extensively in TBS plus 0.5% Tween-20 (TBST) and then incubated for 1 h with IRDye 680RD (Cat# 925-68072; LI-COR) and IRDye 800CW (Cat# 925-32213; LI-COR) secondary antibodies. The membranes were washed extensively with TBST and TBS followed by imaging on a LI-COR Odyssey for chemiluminescent imaging with identical settings for all comparisons.

## Long-term growth and senescence assay

50,000 iBRG1 or iΔHSA cells were plated in individual wells of a 24-well plate and cultured with 1X stock of a 1,000 × 10 $\mu$g/ml doxycycline solution in DMEM for three biological replicates for each cell line and treatment condition. Cells were cultured continuously in either vehicle or doxycycline induction medium and then cultured into larger plates every 72 h. At every culture timepoint, 10 $\mu$l of total cells was used to count in a TC20 Automated Cell Counter (Bio-Rad) and the rest was plated into the larger container. After 12 d of treatment, 350,000 cells from each group were plated in six-well plates, and 48 h later, cells were used to detect the SASP as previously described (Itahana et al, 2007) or cells were collected and total RNA was isolated as described above for qRT-PCR and RNA sequencing. Briefly, cells were fixed using 4% paraformaldehyde for 10 min and washed with PBS. Cells were stained with β-galactosidase staining solution (1 mg/ml X-Gal, 40 mM citric acid/sodium phosphate, pH 6.0, 5 mM potassium ferricyanide, 5 mM

potassium ferrocyanide, 150 mM NaCl, and 2 mM MgCl$_2$) overnight at 37°C in the dark. Cells were then washed two times with PBS and imaged using an AxioVert A1 microscope with an AxioCam 503c camera with identical settings (Zeiss).

## CUT&RUN

CUT&RUN was performed as described with some modifications (Skene & Henikoff, 2017). In detail, 300,000 iBRG1 or iΔHSA cells from plates treated with vehicle or doxycycline for 24 h were collected and linked to concanavalin A beads (Bangs Laboratories). Cells bound to beads were incubated with antibodies overnight at 4°C (1: 50 dilution of anti-V5, #46-0705; Invitrogen), washed with Digitonin wash buffer, and bound with protein-AG fused to MNase by incubation for 1 h. Cells were washed again with Digitonin wash buffer, and then, calcium chloride was added to a final concentration of 2 mM to induce MNase digestion. Digestion was allowed to proceed on ice for 30 min and halted by the addition of 2X stop buffer (340 mM NaCl, 20 mM EDTA, 4 mM EGTA, 50 $\mu$g/ml RNase A, and 50 $\mu$g/ml glycogen). DNA was extracted by the addition of SDS (final concentration 0.1%) and proteinase K (final 10 $\mu$g/ml) and incubating at 55°C for 1 h. The material was extracted with one equivalent volume of phenol/chloroform/isoamyl alcohol 25:24:1 (Thermo Fisher Scientific), precipitated, and washed with 100% ethanol. DNA was resuspended in 150 $\mu$l of water followed by the addition of 75 $\mu$l of AMPure XP beads (Beckmann Agencourt). Beads were mixed and incubated for 5 min, and added to a magnet for 2 min, and the supernatant was collected as size-selected fragments. The DNA was precipitated by adding 700 $\mu$l of ethanol and 1 $\mu$l of 20 mg/ml glycogen and spun at 16,000$g$ for 10 min. The pellet was washed with 70% ethanol and resuspended in 20 $\mu$l of 10 mM Tris–HCl and 1 mM EDTA. 10 ng of DNA was used to generate sequencing libraries using the Accel-NGS 2S Plus DNA library kit (Swift Biosciences) according to the manufacturer's instructions with minimal modifications. The modifications were as follows: library amplification was done for 12 cycles to obtain sufficient material for sequencing. The DNA libraries were analyzed for quality using the Bioanalyzer and sequenced on an Illumina NextSeq 500 with 2 × 75 base pair reads for a targeted sequencing depth of 10 million per sample. Two biological replicates were performed for each antibody, cell line, and treatment condition.

## ATAC-seq

Two replicates were used for all experiments. Cells were collected and subjected to the ATAC-seq protocol described by Buenrostro et al (2013). The cells were treated in 10 mM PIPES, pH 6.8, 100 mM NaCl, 300 mM sucrose, 3 mM MgCl2, and 0.1% Triton X-100 and then treated with the Illumina Tagment DNA TDE1 Enzyme and Buffer Kits for 30 min followed by mixing every 10 min. Libraries were sequenced at the NIEHS Epigenomics Core Facility on a NovaSeq instrument, and reads were trimmed with default parameters.

## Co-immunoprecipitation

iBRG1 or iΔHSA cells collected 24 h after treatment with doxycycline from 10 to 150-mm Nunc dishes (Thermo Fisher Scientific) were

scraped into conical tubes and resuspended in 1% formaldehyde and crosslinked for 10 min at room temperature. Crosslinks were halted by the addition of glycine to a final concentration of 125 mM and incubated for 10 min. Cells were washed with PBS three times and then spun at 600$g$ for 10 min at room temperature. Pelleted cells were resuspended in five cell volumes of high-salt extraction buffer (100 mM Tris, pH 8, 420 mM NaCl, 3 mM MgCl$_2$, 1% NP-40, and 0.1% SDS) for 30 min on ice, followed by five passages through a 25-gauge needle. Pierce Universal Nuclease (Thermo Fisher Scientific) was added at 1:1,000 of the total volume and incubated with rotation for 1 h at 4°C. The lysates were spun at 16,000$g$ for 10 min at 4°C, and the supernatant was collected. Protein concentration was determined using the Bio-Rad Protein Assay, and 2 mg of total protein lysate was diluted 1:5 with IP buffer (100 mM Tris, pH 8, 1 mM EDTA, 100 mM NaCl, and 0.1% NP-40) and 5 $\mu$g of anti-V5 (#46-0705; Invitrogen) was added and rotated overnight at 4°C. The next day, 20 $\mu$l per sample of Protein G Dynabeads (Thermo Fisher Scientific) was washed with IP buffer and resuspended in 20 $\mu$l of IP buffer per sample, and 20 $\mu$l of beads was added to each sample, followed by 3 h of rotation at 4°C. Bead–antibody conjugates were placed on a magnetic stand and then were washed three times with IP wash buffer (50 mM Tris, pH 7.5) by adding 1 ml of IP wash buffer and rotating at 4°C for 5 min followed by placing on a magnetic stand and removal of the supernatant. For Western blots, after the final wash 40 $\mu$l of 2X Laemmli sample buffer (Bio-Rad) was added to the beads and boiled for 10 min, followed by a short spin and addition to a magnet. The supernatant was collected and run on a blot as described above. The following antibodies were used: anti-V5 (Invitrogen), BCL7B (11740-1-AP; Proteintech), BCL7C (PA5-30308; Invitrogen), BAF53a (E3W2A; Cell Signaling Technologies), normal mouse IgG (sc-2025; Santa Cruz Biotechnology), BAF155 (PA5-30174; Invitrogen), GLTSCR1L (PA5-56126; Thermo Fisher Scientific), and BAF45b (PA5-61895; Thermo Fisher Scientific).

For mass spectrometry after immunoprecipitation, after three washes with IP wash buffer, beads were washed two times in 50 mM ammonium bicarbonate and snap-frozen before MS sample preparation. All experiments were done with three biological replicates for each antibody in each condition.

### MS sample preparation

Proteins on beads were digested with 0.5 mg trypsin for 16 h at 37°C. Beads were then heated to 65°C for 10 min and allowed to cool to 37°C, and then, an additional 0.5 mg of trypsin was added, and proteins and peptides were digested for an additional 2 h. The digests were desalted via solid-phase extraction using 100 mg C18 SampliQ cartridges (Agilent) and an SPE apparatus applying head pressure with nitrogen gas. Cartridges were prewashed with 0.1% formic acid in acetonitrile (1 ml) and re-equilibrated with two washes of 0.1% formic acid (1 ml). Peptide digests were then applied to the cartridges. The digests were washed three times (1 ml) with 0.1% formic acid, and peptides were eluted three times (200 ml) with 1:1 (0.2% formic acid:acetonitrile). The elutions were pooled, lyophilized, and then resuspended in 25 $\mu$l 0.1% formic acid.

### LC–MS analyses

Protein digests were analyzed by LC/MS on a Q Exactive Plus mass spectrometer (Thermo Fisher Scientific) interfaced with a nanoACQUITY UPLC system (Waters Corporation) equipped with a 75 $\mu$m × 200 mm HSS T3 C18 column (1.8 $\mu$m particle; Waters Corporation) and a Symmetry C18 trapping column (180 $\mu$m × 20 mm) with 5 $\mu$m particle size at a flow rate of 450 nl/min. The trapping column was positioned in line of the analytical column and upstream of a micro-tee union, which was used both as a vent for trapping and as a liquid junction. Trapping was performed using the initial solvent composition. 5 $\mu$l of the digested sample was injected onto the column. Peptides were eluted by using a linear gradient from 99% solvent A (0.1% formic acid in water [vol/vol]) and 1% solvent B (0.1% formic acid in acetonitrile [vol/vol]) to 40% solvent B over 100 min. For the mass spectrometry, a data-dependent acquisition method was employed with an exclusion time of 15 s and an exclusion of +1 charge states. The mass spectrometer was equipped with a NanoFlex source and was used in the positive ion mode. Instrument parameters were as follows: sheath gas, 0; auxiliary gas, 0; sweep gas, 0; spray voltage, 2.7 kV; capillary temperature, 275°C; S-lens, 60; scan range (m/z), 200–2,000; isolation window, 2 m/z; resolution, 70,000; automated gain control (AGC), 2 × 10$^5$ ions; and a maximum IT, 200 ms. Mass calibration was performed before data acquisition using the Pierce LTQ Velos Positive Ion Calibration mixture (Thermo Fisher Scientific). Peak lists were generated from the LC/MS data using Mascot Distiller (Matrix Science), and the resulting peak lists were searched using the Spectrum Mill software package (Agilent) against the SwissProt database. Searches were performed using trypsin specificity and allowed for one missed cleavage and variable methionine oxidation. Mass tolerances were 20 ppm for MS scans and 50 ppm for MSMS scans.

### Sequential salt fractionation

10 million iBRG1 or iΔHSA cells collected 24 h after treatment with either vehicle or doxycycline were harvested by scraping from 150-mm Nunc dishes (Thermo Fisher Scientific) and spun at 600$g$ for 5 min at room temperature. Cells were resuspended in 1 ml of hypotonic buffer (300 mM sucrose, 60 mM KCl, 60 mM Tris, pH 8, 2 mM EDTA, 0.5% NP-40, and EDTA-free cOmplete protease inhibitor), incubated at 4°C with rotation. Cells were spun at 6,000$g$ for 5 min at 4°C. To the pelleted nuclei, 200 $\mu$l of 0 mM salt buffer (50 mM Tris, pH 8, 1% NP-40, and 1 mM EDTA) was added and the pellet was resuspended and placed on ice for 3 min. The nuclei were spun at 6,000$g$ for 3 min at 4°C, and the supernatant was collected as the 0 mM fraction. The process was repeated with buffers containing 100, 200, 300, 400, 500, and 600 mM NaCl. To each fraction, 70 $\mu$l of NuPAGE LDS sample buffer 4X (Thermo Fisher Scientific) was added and the samples were incubated at 95°C for 10 min and briefly spun. 30 $\mu$l of each sample was added to an 18-well Criterion 4–20% Tris–HCl gel and run at 100 V for 90 min. The gels were transferred to nitrocellulose paper (Bio-Rad) for 2 h at 400 mA at 4°C. The blots were blocked for 1 h at room temperature in TBS containing 5% milk. The membranes were incubated overnight at 4°C with the following primary antibodies at 1:1,000: anti-V5 (#46-0705; Invitrogen), BCL7C (PA5-30308; Invitrogen), BAF53a (E3W2A; Cell Signaling Technologies), and SMARCC1 (PA5-55058; Thermo Fisher Scientific). The membranes were then washed extensively in TBS plus 0.5% Tween-20 (TBST) and then incubated for

1 h with IRDye 680RD (Cat# 925-68072; LI-COR) and IRDye 800CW (Cat# 925-32213; LI-COR) secondary antibodies. The membranes were washed extensively with TBST and then TBS followed by imaging on a LI-COR Odyssey for chemiluminescent imaging with identical settings for all comparisons. To calculate the percentage of protein eluted, the total intensity of all bands was summed and the percentage of the total from each band at each salt concentration was determined. All experiments were performed in duplicate, and the images represent a single biological replicate.

### siRNA transfection

Transfections of siRNAs were carried out with OptiMEM using Lipofectamine 2000 (Invitrogen) according to the manufacturer's instructions with 20 $\mu$M duplex siRNAs of a set of two siRNAs for each gene after initial tests of knockdown by individual siRNAs. 48 h post-transfection, cell media were replaced with DMEM and vehicle or 1X doxycycline was added for iBRG1 or iΔHSA induction. The extent of knockdown was measured by Western blot to detect protein levels and qRT-PCR to detect changes in RNA levels. The sequence information of the siRNAs used in this study from Dharmacon is as follows: *bcl7a* 1: 5′-GGACAUGCAUGACGAUAAC-3′, *bcl7a* 2: 5′-GCCCAAGGUUGAUGACAAA-3′; *bcl7b* 1: 5′-CCGAGAACCUA AUGGCUUU-3′; *bcl7b* 2: 5′-CCUCGGAAGUUGCUGAUGA-3′; *bcl7c* 1: 5′-GAGAAGCGAUGGGUGACUG-3′, *bcl7c* 2: 5′-AAGCUUACCCUGUCUUUGA-3′; and non-targeted control: Dharmacon pool D-001810-10-05.

### Next-generation sequencing data analysis

For RNA-sequencing analysis, adapter sequences were trimmed from reads using Cutadapt (Martin, 2011) and low-quality reads were removed from analysis using Sickle 1.33 (Joshi & Fass, 2011). Reads were aligned to hg19 using STAR (Dobin et al, 2013), gene counts were generated using Salmon (Patro et al, 2017), and differential expression analysis was performed on aggregated gene pseudocounts using limma-voom 3.42.2. DEGs were defined by fold change greater than 1.5 and Benjamini–Hochberg-adjusted *P*-value of less than .05 (Benjamini & Hochberg, 1995; Law et al, 2014).

For pathway analysis, DEGs were tested for gene set enrichment in R version 3.6.1 using clusterProfiler 3.14.3 (Yu et al, 2012) and canonical pathways from MSigDB via the R package msigdbr 7.2.1, with a universe of detected genes for each comparison. Multi-enrichment analysis was performed (Farris et al, 2019) using the top 20 pathways with adjusted *P*-value below 0.1 and two or more DEGs. The gene-pathway incidence matrix was subject to hierarchical clustering and cut into four or five branches to produce a gene-pathway cluster concept network.

For CUT&RUN, adapter sequences were trimmed using Cutadapt (Martin, 2011) and reads were aligned with Bowtie2 (Langmead & Salzberg, 2012). Reads were de-duplicated with Picard (Picard Toolkit, 2019) and processed with SAMtools (Li et al, 2009). Peaks were called using MACS2 (Zhang et al, 2008) with a *P*-value cutoff of .0001 and then filtered to remove peaks from negative controls and to have a minimum score of 5. Coverage files and heatmaps were generated using deepTools (Ramírez et al, 2016). Coverage of individual genes was visualized using Integrative Genomics Viewer (Robinson et al, 2011). Overlaps of peaks and annotation of peaks to genomic regions were performed using ChIPpeakAnno (Zhu et al, 2010) and ChIPseeker (Yu et al, 2015).

### Statistical analysis

Significance of quantitative real-time PCR was determined using two-way ANOVA tests. Error bars represent the SD of three biological replicates.

## Data Availability

All RNA-seq and CUT&RUN data generated for this publication have been deposited in NCBI's Gene Expression Omnibus (Edgar et al, 2002) and are accessible through the GEO Series accession number GSE188730. The mass spectrometry proteomics data have been deposited to the ProteomeXchange Consortium via the PRIDE partner repository with the dataset identifier PXD029647 (Perez-Riverol et al, 2019).

## Supplementary Information

## Acknowledgements

We thank the members of the Archer group and members of the NIEHS Epigenetics and Stem Cell Biology Laboratory for their continued advice and support. We thank Molly Cook, Nicole Reeves, Jason Malphurs, and Greg Solomon of the NIEHS Epigenomics Core Laboratory for their next-generation sequencing expertise and assistance. We also thank Jason Williams of the NIEHS Mass Spectrometry Research and Support Group for help with mass spectrometry data collection and analysis. This work was supported by funding from the National Institute of Environmental Health Sciences (Z01-ES071006-22).

### Author Contributions

N Dietrich: conceptualization, formal analysis, investigation, visualization, methodology, and writing—original draft, review, and editing.
K Trotter: data curation, validation, investigation, and methodology.
JM Ward: software, formal analysis, visualization, methodology, and writing—original draft.
TK Archer: conceptualization, resources, supervision, funding acquisition, investigation, project administration, and writing—original draft, review, and editing.

### Conflict of Interest Statement

The authors declare that they have no conflict of interest.

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
