## [Reviewer comments · Life Science Alliance]

Life Science Alliance

BRG1 HSA domain interactions with BCL7 proteins are critical for remodeling and gene expression

Nicholas Dietrich, Kevin Trotter, James Ward, and Trevor Archer

DOI: <https://doi.org/10.26508/lsa.202201770>

Corresponding author(s): Trevor Archer, National Institutes of Health Clinical Center

Review Timeline:

Submission Date:	2022-10-16
Editorial Decision:	2022-11-13
Revision Received:	2023-01-05
Editorial Decision:	2023-01-27
Revision Received:	2023-02-03
Accepted:	2023-02-06

Scientific Editor: Novella Guidi

Transaction Report:

November 13, 2022

Re: Life Science Alliance manuscript #LSA-2022-01770-T

Dr. Trevor K. Archer
NIH/NIEHS
Epigenetics and Stem Cell Biology Laboratory
NIH, NIEHS
Epigenetics and Stem Cell Biology Laboratory
Research Triangle Park, NC 27709

Dear Dr. Archer,

Thank you for submitting your manuscript entitled "BRG1 HSA domain interactions with BCL7 proteins are critical for remodeling and gene expression" to Life Science Alliance. The manuscript was assessed by expert reviewers, whose comments are appended to this letter. We invite you to submit a revised manuscript addressing the Reviewer comments.

Thank you for this interesting contribution to Life Science Alliance. We are looking forward to receiving your revised manuscript.

Sincerely,

B. MANUSCRIPT ORGANIZATION AND FORMATTING:

Reviewer #1 (Comments to the Authors (Required)):

This manuscript characterizes the effects of the HSA domain of BRG1 on SWI/SNF complex formation and function. In agreement with structural data, the authors find that the HSA is required for the incorporation of BAF53a and BCL7 (and presumably ACTB). Using the SW-13 cell line, the authors characterize how the deletion of HSA domain affects transcription and chromatin remodeling. Interestingly, the authors find that the HSA is required for the majority of SWI/SNF-mediated transcription and accessibility, but not binding. The deletion of BCL7a/b/c transcriptionally mimics the HSA deletion, which is a really interesting and exciting finding. Based on Cryo-EM structures it is a reasonable hypothesis that this arm of the complex may be more involved in modulating the remodeling mechanism of SWI/SNF as opposed to chromatin targeting, but this is the first manuscript I know of that demonstrates this. I look forward to future studies that are designed to figure out how BCL7/BAF53 are actually facilitating this function. It is a really nice study that should be published ASAP. I only have minor issues with the presentation of Figure 3, outlined below:

Figure 3 is confusing, particularly 3A. It would help to split it into more panels and refer to them all in the text with a little more detail. Is 3A middle left mislabeled? It appears that deltaHSA ATAC peaks are bigger than BRG1 ATAC peaks, which doesn't make sense. Some of the labels are cut off as well. In the text, 3B refers to the Cut and Run but that seems to still be ATAC-Seq data in the figure.

Reviewer #2 (Comments to the Authors (Required)):

The SWI/SNF chromatin remodellers are highly important multisubunit complexes that modulate chromatin accessibility. They consist of an ATPase (either BRG1 or BRM), the ARP module comprising BCL7, BAF53, and ACTB and interacting with the HSA domain of the ATPase, and the core module, which is combinatorically composed of multiple subunits. The ARP module is indicated to modulate the ATPase activity and coupling it to DNA translocation. However, the role of the HSA domain of ATPase as well as the ARP module and its components in mammalian cells is poorly understood despite their cancer association. The authors of this study set out to study the impact of the HSA domain of the ATPase BRG1 by using a cancer cell line SW13 that usually does not express functional SWI/SNF complex ATPases and generating genetically modified cell lines that Doxycycline-dependent either express wildtype BRG1 (iBRG1) or BRG1 with a deleted HSA domain (iΔHSA). In contrast to the reconstitution of full-length BRG1, the iΔHSA overexpression hardly affected gene expression. They showed that this differential effect was not due to the different chromatin association of the two different BRG1 proteins and that the HSA domain appears to be dispensable for the binding specification of BRG1. Overexpression of BRG1 without the HSA domain showed less chromatin accessibility changes than reconstitution of the full-length BRG1, indicating an impact on the remodelling function of the SWI/SNF complexes. The authors speculate that this more ineffective remodelling, especially at enhancer regions, may be causative for the differential gene expression alterations. In line with previous reports and recent structural analyses, they show that the HSA domain is required for BCL7 and BAF53 binding to the complexes, which further increases their protein levels (likely due to increased stability) and chromatin association. Finally, using siRNA-mediated knockdown, they show that the BCL7 proteins of the HSA-domain-binding ARP module are important for BRG1-mediated gene expression regulation. So overall, their data strengthen the model that the HSA domain is essential for the interaction of BRG1 with the BCL7 proteins and that this interaction is required for proper BRG1-driven SWI/SNF remodelling activity to control gene expression. However, the exact role of the HSA domain and the impact of its individual binding proteins on the functionality of the BAF complexes remain unsolved.

Major comments

The authors nicely show in a human cell line that the HSA domain of the SWI/SNF ATPase BRG1 is important for effective remodelling activity of the SWI/SNF chromatin complexes rather than defining their chromatin binding specificity. Also in line with current literature, they show that the HSA domain mediates the association of the BCL7 and BAF53 proteins to the complex. Based on literature, these findings are not unexpected and support several previous publications. To increase the novelty of their study, the authors could expand on their last figure to investigate the role of the different HSA-binding proteins (e.g. BCL7A/B/C and BAF53A/B) on the chromatin remodelling activity by single and combinatorial siRNA experiments and measuring chromatin accessibility. A recent BioRxiv paper suggest differential regulatory roles of BAF53 and BCL7 (<https://www.biorxiv.org/content/10.1101/2021.10.26.465931v1>). Furthermore, the authors cannot exclude differential effects due

to the shorter sequence of BRG1 with depleted HSA domain. Using a construct with a random filling sequence could serve as an important control. Finally, the manuscript would increase its impact if the authors would include the investigation of cancer-specific mutations in the HSA domain (e.g. checking their impact on chromatin accessibility/remodelling as well as protein interaction).

A more integrative analysis of their genomics data (RNA-seq, ATAC-seq, Cut&Run) would further provide more value to the paper. The number of DEGs, the overlap of iBRG1 and iΔHSA binding sites, and of the binding sites with the DACs seem low. Overall, their data appears solid and support their main points.

The authors established a nice model system, though it encompasses some disadvantages as the cells do not appreciate the reconstitution of BRG1 and the integration of the overexpression constructs is random. At least showing similar effects in several clones (as the authors indicate to have done), determining the integration site and/or using an alternative validation system would strengthen their manuscript.

Minor comments

Paragraph 1 and 2 of the results section are quite long and could be condensed.

The figures need further polishing, as e.g. labels are sometimes missing, too small or not fully shown. Fig. 2B iHSA instead of iΔHSA.

Figure 3E is missing in the figure, but mentioned in the text.

Figure 4 is not completely novel and could be combined with figure 5.

On page Page 15 the authors conclude that the HSA domain "drives higher affinity interactions with chromatin" of BCL7C and BAF53a. It is likely that these proteins only bind to chromatin via their interaction with the HSA domain of BRG1 in SWI/SNF complexes.

BRM not Brm

Fig. 1B: iΔHSA showed no to minor changes.

Catalog and lot numbers of e.g. antibodies are not all included in the material and method section.

Figure 6A: figure legend for bar graphs is missing.

Figure 6B: single knockdowns as mentioned in the legend are not shown; HSA is not explained in legend.

Referee Cross-Comments

I agree with the other reviewers that the study is well done and that it confirms hypotheses that have already been reported. In agreement with reviewer three, the authors should carefully revise the wording of their conclusions. Overall, I support the publication after further text and figure editing, although the proposed experiments would strengthen the manuscript and their inclusion is recommended.

Reviewer #3 (Comments to the Authors (Required)):

Here Dietrich et al. report their results on the impact of BRG1 HSA domain deletion on BRG1 chromatin association (CUT&RUN), chromatin remodeling (ATAC-seq), and BAF complex stability (IP-MS) and chromatin association (SSE). This nice set of data support that the HSA domain is important for transcriptional regulation of a subset of BRG1 target genes, BAF remodeling activity, as well as stabilization of the interaction between the BCL7C and BRG1. While recently several structures of BAF complexes have been solved the relative importance of all of the subunit interactions in complex activity largely remain a mystery. As BAF is heavily mutated in cancer, it is of great interest to understand how mis-regulated complexes are functioning and thus this study is timely. Overall, it is well reasoned, nicely carried out, and should be of strong interest to the field. I have only a few minor comments:

1) In the following sentence at the end of page 7 "The results demonstrated here support a model in which re-expression of BRG1 in cells without any apparent SWI/SNF ATPase and remodeling function..." is unclear. In particular, it is unclear where the statement without apparent activity is coming from. Please clarify.

2) On page 12 it is noted that the iDeltaHSA allows for ATPase function. I would argue that they observe changes in chromatin accessibility still, but as ATPase activity is not being directly tested this should be reworded.

3) Similar to point 2, at the end of page 14 it is stated that the HSA is critical for binding BCL7A, B, and C but it was only directly observed for C. In addition, it is noted that loss of BCL7 alters subsequent remodeling activity. Though it is correlated it is not a direct measure. Please re-word.

4) Regarding Figure 5B BCL7C results - first, it is stated that HSA increases BCL7C affinity for chromatin, but this suggests direct affinity and results may just be due to increased complex stability and thus retention of BCL7C at chromatin. Second, it should not be stated that there is a shift in the elution as BCL7C upon deletion of HSA as it not even observed in iDeltaHSA.

5) There are a few typos throughout as well as very long paragraphs that sometimes made the flow difficult.

Re: Life Science Alliance manuscript #LSA-2022-01770-T**Reviewer #1:**

Issue #1: *Figure 3 is confusing, particularly 3A. It would help to split it into more panels and refer to them all in the text with a little more detail. Is 3A middle left mislabeled? It appears that deltaHSA ATAC peaks are bigger than BRG1 ATAC peaks, which doesn't make sense.*

Response: We agree that the presentation of Figure 3A could be improved. We separated Figure 3A into multiple panels and referred to them in the text individually to provide additional clarity. The label for 3A middle, which is now Figure 3C, was improved and the difference in size is expanded on within the results section of the text. The observed phenomenon is largely due to those peaks having the strongest signal and being the overlapping peaks between *iBRG1* and *iΔHSA*. In addition, we separated out the defined unique peaks for each to demonstrate these differences in the figure.

Issue #2: *Some of the labels are cut off as well.*

Response: We thank the reviewer for identifying some formatting issues and we have addressed these problems by moving the panels within the figures to include the labels on all figures.

Issue #3: *In the text, 3B refers to the Cut and Run but that seems to still be ATAC-Seq data in the figure.*

Response: We thank the reviewer for highlighting this issue and we have corrected the reference in the text and figure legend to separate the references to the Cut&Run data versus the ATAC-Seq data.

Reviewer #2

Issue #1: *To increase the novelty of their study, the authors could expand on their last figure to investigate the role of the different HSA-binding proteins (e.g. BCL7A/B/C and BAF53A(/B)) on the chromatin remodelling activity by single and combinatorial siRNA experiments and measuring chromatin accessibility.*

Response: We thank the reviewer for this suggestion, and we agree that this will be a valuable addition to the field. However, we believe that these additions would instead be an integral part an additional study that is focused on specific functions of the *BCL7* and *BAF53* proteins and as such, are beyond the scope of this manuscript. We feel that the focus of this study is the BRG1-HSA domain, and

the characterization of this domain has led us to new avenues to elucidate further functions of these proteins independently in the future.

Issue #2: *The authors cannot exclude differential effects due to the shorter sequence of BRG1 with depleted HSA domain. Using a construct with a random filling sequence could serve as an important control. Finally, the manuscript would increase its impact if the authors would include the investigation of cancer-specific mutations in the HSA domain (e.g. checking their impact on chromatin accessibility/remodeling as well as protein interaction).*

Response: We thank the reviewer for these excellent suggestions and agree that the deletion of an entire domain could affect the structure and function of *BRG1*. We feel that the ability of the *HSA* deletion to still form a protein that forms a *SWI/SNF* complex suggests that the shortening of *BRG1* is not the dominant cause of the phenotype we describe. While the introduction of a sequence within this domain could likely provide a strong negative control, we feel that the design and characterization of this domain, that does not negatively affect the folding and structure of *BRG1*, would provide additional complications to the system that might not reflect the normal biology of *BRG1*. We also feel that examining multiple individual mutations within the domain would be a valuable addition to the field, and indeed is an interest of the laboratory. However, the scale and scope of such experiments but would be outside the scope of this manuscript.

Issue #3: *A more integrative analysis of their genomics data (RNA-seq, ATAC-seq, Cut&Run) would further provide more value to the paper. The number of DEGs, the overlap of *iBRG1* and *iΔHSA* binding sites, and of the binding sites with the DACs seem low. Overall, their data appears solid and support their main points.*

Response: We thank the reviewer for this comment, recognizing the quality of the data, and while the overlap between datasets is low, we feel that the data overall reflects the differences between the wild-type and mutant *BRG1* proteins. Specifically, the DEGs we identified are likely the result of indirect binding and accessibility changes, likely at distal enhancers, and we have highlighted this observation within the revised text in the manuscript.

Issue #4: *The authors established a nice model system, though it encompasses some disadvantages as the cells do not appreciate the reconstitution of BRG1 and the integration of the overexpression constructs is random. At least showing similar effects in several clones (as the authors indicate to have done), determining the integration site and/or using an alternative validation system would strengthen their manuscript.*

Response: To address this reviewer question, we have included new data from two additional *iΔHSA* clonal cell lines that display the same phenotype as the clone used throughout the manuscript, which can be seen in **Supplementary Figure 1** and is highlighted in the revised results section of the manuscript.

Issue #5: *Paragraph 1 and 2 of the results section are quite long and could be condensed.*

Response: We agree that these two paragraphs were too expansive, and we have reduced the length of the text while maintaining the overall description of the results.

Issue #6: *The figures need further polishing, as e.g. labels are sometimes missing*

Response: We agree and thank the reviewer. Please see the response to Reviewer #1, Issue #2 above.

Issue #7: *Fig. 2B iHSA instead of iΔHSA*

Response: We have fixed this issue within the Figure and addressed any possible issues within the text.

Issue #8: *Figure 3E is missing in the figure, but mentioned in the text*

Response: We thank the reviewer for identifying this error, we have corrected the figure to reflect the correct references within the text. Please see Reviewer #1 Issue #1 for further explanation of alterations to this figure.

Issue #9: *Figure 4 is not completely novel and could be combined with figure 5.*

Response: We agree with the reviewer that the data presented within Figure 4 is not novel enough to require presentation on its own, and due to this suggestion, we have merged Figure 4 and Figure 5 and reworded the text to reflect this change.

Issue #10: On page Page 15 the authors conclude that the HSA domain "drives higher affinity interactions with chromatin" of BCL7C and BAF53a. It is likely that these proteins only bind to chromatin via their interaction with the HSA domain of BRG1 in SWI/SNF complexes.

Response: We agree with the reviewer and have reworded the text to emphasize that the HSA domain is needed for the interaction of BCL7C and BAF53 with the

rest of the SWI/SNF complex rather than the HSA domain being responsible for direct binding to other complex members.

Issue #11: *BRM not Brm*

Response: We thank the reviewer for this correction, and we have edited all references of Brm in the text to fix this issue.

Issue #12: *Fig. 1B: Δ HSA showed no to minor changes.*

Response: We agree with the reviewer regarding the wording of this result and have addressed this issue within the revised text to reflect this observation.

Issue #13: *Catalog and lot numbers of e.g. antibodies are not all included in the material and method section.*

Response: We thank the reviewer for identifying this error, we have included catalog numbers for all antibodies directly in the materials and methods.

Issue #14: *Figure 6A: figure legend for bar graphs is missing.*

Response: We thank the reviewer for this observation, and we have included the legend for this figure in the figure legends section of the text.

Issue #15: *Figure 6B: single knockdowns as mentioned in the legend are not shown; HSA is not explained in legend.*

Response: To address this reviewer's comment we have removed the reference to the single knockdowns in the legend. We have fixed the reference to *HSA* as it should have instead been worded as " *Δ HSA*".

Reviewer #3

Issue #1: *In the following sentence at the end of page 7 "The results demonstrated here support a model in which re-expression of BRG1 in cells without any apparent SWI/SNF ATPase and remodeling function..." is unclear. In particular, it is unclear where the statement without apparent activity is coming from. Please clarify.*

Response: We agree with the reviewer that the statement regarding the SWI/SNF activity was unclear. To address this issue, we have expanded on the explanation that neither *BRG1* nor *Brm* is expressed within the SW-13 cell line within the text. The lack of these ATPase proteins is the point that we were trying to make, and we have reworded the text to be clearer.

Issue #2: *On page 12 it is noted that the $i\Delta HSA$ allows for ATPase function. I would argue that they observe changes in chromatin accessibility still, but as ATPase activity is not being directly tested this should be reworded.*

Response: To address this reviewer issue, we have reworded this text to highlight that we are focused on chromatin accessibility rather than directly testing ATPase function in any of our assays. We have removed and edited wording throughout to reinforce this difference in the assay that is being performed.

Issue #3: Similar to point 2, at the end of page 14 it is stated that the HSA is critical for binding BCL7A, B, and C but it was only directly observed for C. In addition, it is noted that loss of BCL7 alters subsequent remodeling activity. Though it is correlated it is not a direct measure. Please re-word.

Response: We agree with the reviewer that we did not directly observe binding to BCL7A or BCL7B and have specifically highlighted that we only see binding of iBRG1 to BCL7C. We have also removed the reference to the remodeling function driven by BCL7C loss, as the reviewer correctly points out that it is a correlation instead of a direct measurement. Instead, we have highlighted that this is a predicted result that should be tested in future studies.

Issue #4: *Regarding Figure 5B BCL7C results - first, it is stated that HSA increases BCL7C affinity for chromatin, but this suggests direct affinity and results may just be due to increased complex stability and thus retention of BCL7C at chromatin. Second, it should not be stated that there is a shift in the elution as BCL7C upon deletion of HSA as it not even observed in $i\Delta HSA$.*

Response: We agree with the reviewer that the increased complex stability and retention of BCL7C at chromatin is an alternative explanation for the results shown in **Figure 5**, which has been reformatted to be **Figure 4**. We have directly addressed this possibility in the text and de-emphasized the possibility of direct affinity, which we did not measure in this assay. We also reworded the text addressing the salt extraction results for this BCL7C in $i\Delta HSA$ cells to reflect that we do not observe BCL7C, instead we explain that “There was little to no detection of BCL7C expression or elution in $i\Delta HSA$ expressing cells”.

Issue #5: *There are a few typos throughout as well as very long paragraphs that sometimes made the flow difficult.*

Response: We thank the reviewer for identifying these errors and apologize. We have thoroughly examined and corrected typos as well as reduced the length of paragraphs to improve the readability of the text.

January 27, 2023

RE: Life Science Alliance Manuscript #LSA-2022-01770-TR

Dr. Trevor K. Archer
National Institutes of Health
Epigenetics and Stem Cell Biology Laboratory
NIH, NIEHS
Epigenetics and Stem Cell Biology Laboratory
Research Triangle Park, NC 27709

Dear Dr. Archer,

Thank you for submitting your revised manuscript entitled "BRG1 HSA domain interactions with BCL7 proteins are critical for remodeling and gene expression". We would be happy to publish your paper in Life Science Alliance pending final revisions necessary to meet our formatting guidelines.

- please address the final Reviewer 2's comments
- please add ORCID ID for corresponding author-you should have received instructions on how to do so
- please use the [10 author names, et al.] format in your references (i.e. limit the author names to the first 10)
- please double-check your callouts and add a figure number to the callout on page 16; (Figure I callout should be Figure 4I and Figure 5J should be Figure 4J)
- please add a figure callout for Supplemental Figure5 in your main manuscript text

Figure Check:

- please add scale bars to Figure S2B and Figure S3
- Figure 4D: looks like a splice before the last column in both rows. Please provide source data for this figure panel

A. FINAL FILES:

B. MANUSCRIPT ORGANIZATION AND FORMATTING:

Sincerely,

Reviewer #2 (Comments to the Authors (Required)):

While the authors added few new data, they did adjust the text and figures to address some of the reviewers' comments and concerns. However, some of the claimed changes have actually not been incorporated (e.g., BRM), and there remain other issues with the quality of the figures and text that should be resolved as part of the editorial process. Overall, I support the publication from a scientific point of view and would appreciate it if the authors would add a sentence about the limitations of their system.

Reviewer #3 (Comments to the Authors (Required)):

The authors have addressed all of my previous concerns and I believe this is suitable for publication.

Editorial Review:

Issue #1: *please address the final Reviewer 2's comments*

Response: Please see Reviewer 2 comment responses below

Issue #2: *please add ORCID ID for corresponding author-you should have received instructions on how to do so*

Response: ORCID ID will be added for final submission.

Issue #3: *please use the [10 author names, et al.] format in your references (i.e. limit the author names to the first 10)*

Response: Reference format has been edited to fit to the guidelines described.

Issue #4: *please double-check your callouts and add a figure number to the callout on page 16; (Figure I callout should be Figure 4I and Figure 5J should be Figure 4J)*

Response: Figure number callouts have been checked for correctness and the specific callouts have been address in the text.

Issue #5: *please add a figure callout for Supplemental Figure5 in your main manuscript text*

Response: Callout has been added to main manuscript text.

Figure Check:

Issue #6: *please add scale bars to Figure S2B and Figure S3*

Response: Scale bars have been added to the requested figures.

Issue #7: *Figure 4D: looks like a splice before the last column in both rows. Please provide source data for this figure panel*

Response: Source data file has been provided. The splice is due to a lane in the original blot for a separate antibody that was found to be non-specific for the protein of interest and was not described in the text.

Reviewer #2

Issue #1: *While the authors added few new data, they did adjust the text and figures to address some of the reviewers' comments and concerns. However, some of the claimed changes have actually not been incorporated (e.g., BRM), and there remain other issues with the quality of the figures and text that should be resolved as part of the editorial process. Overall, I support the publication from a scientific point of view and would appreciate it if the authors would add a sentence about the limitations of their system.*

Response: We appreciate the reviewer's suggestion and have edited any references to BRM to reflect the correct nomenclature. To address the issue of the limitations of the system, the following sentences have been added to the final paragraph of the Discussion section:

"A limitation of our study is that we have not biochemically examined remodeling function, but examined the changes in accessibility driven by either iBRG1 or iΔHSA expression by ATAC-seq. In addition, we cannot completely exclude how a shortened BRG1 protein could drive some of the phenotypes observed here."

“However, an additional limitation of this study is that we have not directly addressed the mechanism and function of each specific BCL7 protein but focused on the knockdown of all three proteins. In addition, we have not directly characterized how the loss of these proteins alters chromatin remodeling or accessibility.”

February 6, 2023

RE: Life Science Alliance Manuscript #LSA-2022-01770-TRR

Dr. Trevor K Archer
National Institutes of Health Clinical Center
11 TW Alexander
RTP 27709

Dear Dr. Archer,

Thank you for submitting your Research Article entitled "BRG1 HSA domain interactions with BCL7 proteins are critical for remodeling and gene expression". It is a pleasure to let you know that your manuscript is now accepted for publication in Life Science Alliance. Congratulations on this interesting work.

DISTRIBUTION OF MATERIALS:

Again, congratulations on a very nice paper. I hope you found the review process to be constructive and are pleased with how the manuscript was handled editorially. We look forward to future exciting submissions from your lab.

Sincerely,
